# COPS in Action: Exploring Structure in the Usage of the Youth Psychotherapy MATCH

**Thomas Rusch** [1,*] **, Katherine Venturo-Conerly** [2] **, Gioia Baja** [3] **and Patrick Mair** [2]

1 Competence Center for Empirical Research Methods, WU Vienna University of Economics and Business, 1020 Vienna, Austria
2 Department of Psychology, Harvard University, Cambridge, MA 02138, USA; kventuroconerly@g.harvard.edu (K.V.-C.); mair@fas.harvard.edu (P.M.)
3 New Yorker, 1110 Vienna, Austria
* Correspondence: thomas.rusch@wu.ac.at

**Abstract:** This article is an introduction to Cluster Optimized Proximity Scaling (COPS) aimed at practitioners, as well as a tutorial on the usage of the corresponding R package **cops**. COPS is a variant of multidimensional scaling (MDS) that aims at providing a clustered configuration while still representing multivariate dissimilarities faithfully. It subsumes most popular MDS versions as special cases. We illustrate the ideas, use, flexibility and versatility of the method and the package with data from clinical psychology on how modules of the Modular Approach to Therapy for Children (MATCH) are used by clinicians in the wild. We supplement the COPS analyses with density-based hierarchical clustering in the original space and faceting with support vector machines. We find that scaling with COPS gives a sensible and insightful spatial arrangement of the modules, allows easy identification of clusters of modules and provides clear facets of modules corresponding to the MATCH protocols. In that respect COPS works better than both standard MDS and clustering.

**Keywords:** clustering; exploratory data analysis; facet theory; ordination; psychotherapy; multidimensional scaling; proximity scaling; youth psychotherapy





## 1. Introduction

Proximity scaling techniques such as multidimensional scaling (MDS) are essential statistical/psychometric tools to explore data sets. Flexible modern approaches such as Cluster Optimized Proximity Scaling (COPS) [1] or Structure Optimized Proximity Scaling (STOPS) [2] allow users to emphasize various structural aspects of the data during the scaling process. Technically, this is achieved by augmenting the loss function that is subject to minimization with structural considerations. For instance, the COPS framework, which we will use in this paper, incorporates a penalty and parametrized transformations with the aim of increasing the clusteredness of the final result. That is, the resulting configuration exhibits improved clustering of the scaled objects, which may give practitioners important structural insights.

The aim of this paper, targeted to applied researchers, is twofold. First, it provides a lightweight introduction to the COPS framework. With a real-world data set, we highlight the applicability and usage of the framework. We include practical suggestions related to the application of COPS and related techniques, and how to turn the different gauges to achieve the best results. The article also functions as a software tutorial on the use of the newly developed R package **cops**; we use it in a stand-alone fashion and in conjunction with complementary R packages, in that this article serves as a blueprint for the typical setting for which we believe the COPS framework to be particularly useful. The tutorial instructions are meant to support practitioners with structural explorations of their own data sets when they utilize the COPS framework.

Second, the paper also has a substantive motivation: Several past reviews of psychotherapies have grouped psychotherapy treatment techniques according to theoretical similarities and differences. These previous expert reviews have identified important domains on which psychotherapy treatment techniques differ such as a focus on insights or on direct behavior change, diagnoses targeted, degree of focus on goals and solutions, clinician directiveness or client directedness, focus on cognitions, behaviors, or emotions, duration and intensity, and individual or family-focus [3–7]. What has been lacking are empirical investigations based on practical usage. The present article addresses that by contributing an empirical investigation into grouping treatment techniques via structural exploration of the modules of the Modular Approach to Therapy for Children with Anxiety, Depression, Trauma, or Conduct Problems (MATCH-ADTC; [8]) based on their use by practicing clinicians, which has relevance for psychotherapy research and clinical psychology.

MATCH-ADTC is a flexible youth psychotherapy designed to suit the needs of youth seeking mental health care in community clinics, most of whom have multiple primary and secondary diagnoses and changing needs throughout treatment. MATCH is a modular youth psychotherapy, meaning that it consists of multiple, unique component parts (i.e., modules), each with a specific function, and that these component parts may be flexibly rearranged, repeated, and omitted according to the unique needs of each client [9]. Each MATCH module (e.g., psycho-education about anxiety for youth and caregivers; rewards for desired behaviors) is drawn from reviews of evidence-based youth psychotherapies [10]. The MATCH manual includes assessment recommendations, flowcharts, and narrative text meant to help clinicians select appropriate module sequences for each client, yet following these recommendations and selecting modules involves the use of clinical judgment; therefore, treatment courses for each client who receives MATCH can be rather unique, even among youths with similar diagnoses.

Using the COPS framework and related statistical methods, we seek to empirically establish the groupings of MATCH modules and explore their relation based on the real use of practitioners in communities. To that end, COPS as a variant of MDS will be used together with a density-based hierarchical clustering method OPTICS [11] as well as the derived goodness-of-clustering summary statistic, the OPTICS Cordillera [12], to scale and explore the usage of the modules of the MATCH-ADTC. We also connect this to facet theory [13] via the SVM-MDS framework of [14] that generates facets with support vector machines (SVM). These substantive results are of interest not only to practitioners and developers of MATCH, but as the first empirical investigation into grouping treatment techniques they can also be interpreted in the context of previous theoretical research, and may be used to guide future research on the effects and use of treatment elements. In doing so, we give an example of how the COPS framework and the software may typically be used. The R code to fully reproduce the results presented in this paper is given as Supplementary Materials.

## 2. Materials and Methods

In this section, we give a description of the data and an overview of the statistical methods and ideas that we use in this study without going into too much technical detail. The main R software package that we use is **cops** [15] for MDS and COPS-related functionality. We also use **dbscan** [16] and **cordillera** [17] for functions related to OPTICS clustering, **smacof** [18] for faceting and MDS infrastructure and **e1071** [19] for fitting SVM.

### 2.1. Multidimensional Scaling (MDS)

The starting point of our methodological elaborations is multidimensional scaling (MDS), a well-known technique for scaling objects in a low-dimensional space [20–24]. The input is a symmetric pairwise dissimilarity matrix $\Delta$ of dimension $o \times o$ with elements $\delta_{ij}$ between object $i$ and object $j$, where $o$ is the number of objects. The main output is a configuration matrix $\mathbf{X}$, typically subject to plotting. $\mathbf{X}$ is of dimension $o \times p$, where $p$ is the number of dimensions, $x_i = (x_{i1}, \ldots, x_{ip})$ is the $i$-th row vector of $\mathbf{X}$.

The most popular approach to find the solution is by minimizing the *stress* target function [24]:

$$\sigma(\mathbf{X}) = \sum_{i<j} (\hat{\delta}_{ij} - d_{ij}(\mathbf{X}))^2 \qquad (1)$$

Here, $d_{ij}(\mathbf{X})$ are the fitted distances between the points in the $p$-dimensional space (almost always Euclidean), and the $\hat{\delta}_{ij}$'s (disparities) are transformed versions of the input dissimilarities $\delta_{ij}$. Classical transformations include (under the restriction $\hat{\delta}_{ij} \geq 0$)

- Ratio transformation $\hat{\delta}_{ij} = b\delta_{ij}$;
- Interval transformation: $\hat{\delta}_{ij} = a + b\delta_{ij}$.

Thus, stress-based MDS tries to find a configuration $\mathbf{X}$ whose point-wise distances are as close as possible to a transformed version of the input dissimilarities. Naturally, the smaller the stress, the better the fit [25].

Over the years, various extensions and modifications of Kruskal's original stress formulation have been proposed. They extend the family of MDS methods towards a more general proximity scaling (PS) framework [1]. In order to incorporate some of the extensions relevant in this article, we re-write Equation (1) as follows:

$$\sigma_2(\mathbf{X}) = \sum_{i<j} \left( \hat{\delta}_{ij} - \hat{d}_{ij}(\mathbf{X}) \right)^2. \qquad (2)$$

This reformulation allows us to include transformations for the disparities as well as for the fitted distances $\hat{d}_{ij}(\mathbf{X})$. We may think of

- Logarithmic transformation: $\hat{\delta}_{ij} = \log(\delta_{ij})$ and $\hat{d}_{ij}(\mathbf{X}) = \log(d_{ij}(\mathbf{X}))$;
- Quadratic transformation: $\hat{\delta}_{ij} = \delta_{ij}^2$ and $\hat{d}_{ij}(\mathbf{X}) = d_{ij}(\mathbf{X})^2$.

The logarithmic transformation is also known as the multiscale approach [26]. The quadratic transformation is also known as s-stress [27].

Such transformations allow different values of $\delta_{ij}$ and $d_{ij}(\mathbf{X})$ to receive more or less emphasis compared to classical MDS, for example to emphasize larger values more strongly (as in s-stress) or to express a probabilistic model for the $\delta_{ij}$ (as in multiscale). Within a general proximity scaling framework, the systematic flexibility given by such transformations is of special importance.

One very general and flexible formulation is the *p-stress* (power stress) target function [1,28]

$$\sigma_p(\mathbf{X}) = \sum_{i<j} (b\delta_{ij}^\lambda - d_{ij}(\mathbf{X})^\kappa)^2, \qquad (3)$$

with power coefficients $\lambda$ for the dissimilarities, $\kappa$ for the fitted distances and $b$ as a ratio transformation parameter (so that $b\delta_{ij}^\lambda \geq 0$). This p-stress encompasses most of the popular metric stress functions proposed in the literature [1] (there also exists a version with weights but we omitted the weights for simplicity). Concerning the mentioned models, p-stress gives ratio MDS if $\kappa = \lambda = 1$, s-stress if $\kappa = 2, \lambda = 2$, and approximates multiscale stress with $\kappa \to 0$ if we use $\hat{\delta}_{ij} = \log(\delta_{ij})$ and $\lambda = 1$ (this is due to the identity $\log(d_{ij}(\mathbf{X})) \approx ad_{ij}(\mathbf{X})^{1/a} - a$ for large $a$, that additive and multiplicative constants are irrelevant for the Euclidean distance and that $d_{ij}(\mathbf{X}) \geq 0$).

In p-stress and its special cases, the exponents have the following effects: For $\lambda$, larger values emphasize larger dissimilarities and smaller values emphasize the local structure (smaller dissimilarities). For $\kappa > 1$, larger values mean we map the higher values of $\hat{\delta}_{ij}$ to an increasingly tighter range of $d_{ij}(\mathbf{X})$ (as in a convex function). The differentiation between the large $\hat{\delta}_{ij}$ is down weighted as compared to the smaller $\hat{\delta}_{ij}$ and thus the fitted $d_{ij}(\mathbf{X})$ discriminate more finely between the lower $\hat{\delta}_{ij}$, emphasizing local variation in the $\hat{\delta}_{ij}$. For $\kappa < 1$, smaller values mean we map the lower values of $\hat{\delta}_{ij}$ to a tighter range of

$d_{ij}(\mathbf{X})$ (concave function) and thus the $d_{ij}(\mathbf{X})$ discriminate more finely between the higher $\delta_{ij}$, which emphasizes the global picture (as if zooming out).

In this article, we are interested in scaling the modules of MATCH-ADTC based on their usage by clinicians in the wild. The configurations that we obtain give us insight into how the usage of the modules is structured over the patients. We do this by interpreting the pairwise distances between the points in the configuration plot as the best representation of the higher dimensional proximities. The closer two points, the less dissimilar the corresponding objects are, assuming that the MDS solution fits well. This is one part of the story we tell here: scaling the modules in a lower dimensional space and interpreting their spatial arrangement.

### 2.2. Distance Measure

To apply the methods we describe here, we need a measure of dissimilarity. The raw data are counts (see Section 2.6 for details), so we use a dissimilarity for count vectors (or count profiles, i.e., elements are non-negative), which we call the $\phi$-distance, $d^\phi$. For two non-zero vectors, $x_i = (x_{i1}, \ldots, x_{il}, \ldots, x_{io})$ and $x_j = (x_{j1}, \ldots, x_{jl}, \ldots x_{jo})$,

$$d_{ij}^\phi = d^\phi(x_i, x_j) = \sqrt{\sum_{l=1}^{o} \frac{(x_{il} - x_{jl})^2}{(x_{il} + x_{jl})(n_i + n_j)}} = \frac{1}{\sqrt{N}} \sqrt{\sum_{l=1}^{o} \frac{(x_{il} - x_{jl})^2}{(x_{il} + x_{jl})}} \tag{4}$$

with $N = n_i + n_j$ and $n_k = \sum_l x_{kl}$. Note that in the last expression, the term involving the $x_i, x_j$ is a blended $\chi^2$-distance [29]. Therefore, $d^\phi$ is a blended $\chi^2$-distance between two count vectors normalized with the square root of the overall sum $N$ of the elements in both vectors, similar in spirit to the mean square contingency $\phi^2 = \chi^2/N$ [30]. Note that in case of both vectors being zero vectors, we would have a division by 0 in Equation (4) and the distance is not defined. If only one vector is a zero vector, $d^\phi$ can be calculated, but we believe it only makes sense for non-zero vectors.

The $\phi$-distance has a number of properties that we deem favorable for this situation: Since it is based on a blended $\chi^2$-distance (essentially a re-weighted version), it retains properties of the blended $\chi^2$-distance including that it is a metric, a weighted Euclidean distance and measures mostly the difference in shape of the vector profile. It is also robust [29]. Each $l$-th element is exchangeable between $i, j$, which means we can look at every $l$'s contribution to the distance in isolation. Additionally, the $\phi$-distance only uses the elements of the two vectors $i, j$, for which we want the distance; values in other vectors have no bearing on the distance calculation, making it invariant to the inclusion or exclusion of other vectors. Weighting with $N$ makes the distance lie between 0 (same profile) and 1 (non-overlapping profile, i.e., every element in $x_j$ corresponding to a non-zero element in $x_i$ is zero). Furthermore, if we multiply all elements in both $x_i$ and $x_j$ with a constant, the $\phi$-distance does not change. Conversely, if for a pair of vectors the blended $\chi^2$-distance is equal to another pair, but the latter pair has higher $N = n_i + n_j$, then the $\phi$-distance between the first pair is larger than between the second pair. It can thus account for prevalence differences in the vectors and that larger $\chi^2$-distances between $x_i$ and $x_j$ are randomly more likely if we have larger $n_i$ and $n_j$.

### 2.3. Ordering Points to Identify the Clustering Structure (OPTICS) and the OPTICS Cordillera

As part of this study, we are also interested in identifying data-driven groups of modules based on their usage. This is the task of clustering, which is in many ways complementary to the scaling approach.

Out of the myriad of cluster algorithms, we will use OPTICS, a hierarchical cluster algorithm that has a density-based definition of clustering [11]. Informally, this means that within a neighborhood around each point $x_i$ other points are considered as belonging to the same cluster if the density of points in that area is higher than outside of that area. Thus, a cluster is defined as a dense accumulation of points; clusters are separated from each other

by regions that are sparsely (or not at all) populated by points. The main attractions of this clustering algorithm are that the shape of the accumulation can be anything (as opposed to, say, the Voronoi tessellation of k-means), that the algorithm can find nested clusters, that points that fall into the sparse regions are considered as noise points and that we do not have to specify the number of clusters.

OPTICS needs to make only one assumption about what constitutes a cluster: the minimum number $k$ of objects that must make up a cluster (argument `minPts` in the R function `optics` in **dbscan**). It also features a parameter $\epsilon$ (argument `eps` in `optics`) that defines the neighborhood radius in which to look for points and controls how likely a point is considered as noise; for our purpose it just needs to be sufficiently large as we do not think any of our observations may be noise. Once these two parameters are fixed, the OPTICS algorithm encodes the complete density-based clustering information in the data as an ordering of the points in data matrix $\mathbf{X}$, $R(\mathbf{X})$, and an associated minimum reachability $r_i^*$ of each point $x_i$. This minimum reachability distance can be thought of as a type of single linkage distance of a point $x_i$ to the point $x_j$ that comes before it in the ordering $R(\mathbf{X})$. The ordering itself is so that points that are close in the ordering and have relatively small associated values of $r_i^*$ are an accumulation, whereas points that are far from each other in the ordering, or are close but have relatively large $r_i^*$ can be considered to not be part of the same accumulation.

This combination of $R(\mathbf{X})$ and the set of minimum reachabilities can be visualized in a reachability plot, with the OPTICS ordering of the points on the $x$-axis and each point's $r_i^*$ on the $y$-axis. This plot is a representation of the overall clustering structure of the data. OPTICS does not assign clusters per se, but they can result from interpreting the reachability plot (much like in a dendrogram). The interpretation usually is that a "valley" stands for a possible cluster (as it is a dense accumulation of points) whereas "peaks" are separators between possible clusters (as the minimum reachability of the point that shows the peak to the preceding point is large). The number of valleys can be interpreted as corresponding to the number of clusters. A valley within a valley is a denser accumulation nested within a less dense accumulation. There also is a method, the $\xi$-strategy, to detect and extract clusters hierarchically based on the steepness in the reachability plot and to assign observations to the clusters automatically. In this the $\xi$-parameter can be interpreted as the steepness threshold to identify clusters in the reachability plot. It classifies clusters by the change in relative cluster density. The technical details for all of this can be found in [11].

Based on OPTICS, there also exists a *goodness-of-clustering* index that summarizes the reachability plot into a single number: The OPTICS Cordillera, $OC(\mathbf{X})$ [12]. It measures how rugged the reachability plot is by calculating the length of the up and down of the $r_i^*$ over the ordering $R(\mathbf{X})$. A larger $OC(\mathbf{X})$ stands for a more rugged reachability plot and thus more density-based clusteredness of $\mathbf{X}$. The $OC(\mathbf{X})$ features the same two parameters $k$ and $\epsilon$ with the same role as OPTICS (in the function `cordillera` of R package **cordillera** they are called `minpts` and `epsilon`, respectively). The length can be any $L^q$-norm, with $q \geq 1$ (typically 1 or 2, argument `q` in `cordillera`). The $OC(\mathbf{X})$ has favorable properties for measuring density-based clusteredness in a data matrix, including a monotonic increase in the value if the clusters become more dense, if the clusters become more separated or if the numbers of clusters increases. See [12] for technical details.

We use the normalized version $OC'(\mathbf{X})$, which lies between 0 and 1 with 0 being a perfectly regular arrangement and 1 a perfectly clustered arrangement, both as defined in [12]. For this normalization next to the OPTICS parameters $k$ and $\epsilon$ and the $q$, one also needs to fix a $d_{max}$ (argument `dmax`), which defines the maximal between-cluster-distance to derive the maximally clustered arrangement (maximal clusteredness is defined as having $o/k$ clusters of exactly $k$ objects all coinciding at the same point, respectively, and that these clusters are all at a distance of $d_{max}$ from each other). The $d_{max}$ is also a winsorization limit of the $r_i^*$ so that any $r_i^* > d_{max}$ is set to $d_{max}$; it provides robustness against outliers.

*2.4. Cluster Optimized Proximity Scaling (COPS)*

A standard but ad hoc approach is to use different flavors of MDS to scale the data and visualize the configuration and then use clustering methods to assess the clusteredness and clustering structure of the data in the original space or in the configuration (e.g., [31]). More elegantly, we can also combine MDS and the OPTICS Cordillera in one go to a variant of MDS that is called Cluster Optimized Proximity Scaling or COPS [1]. It is a proximity scaling method that penalizes the MDS fit measure stress with the $OC'(\mathbf{X})$ to allow finding a configuration that is close to the corresponding unpenalized optimal MDS configuration, but shows higher clusteredness. This can be useful for exploring or generating discrete structures or to preserve clusters.

In [1], two ways of how COPS can be used are presented: In one variant (COPS-C), one looks for an optimal configuration $\mathbf{X}^*$ directly, given transformation parameters $\theta$ (e.g., $\theta = (\kappa, \lambda)$ in Equation (3)). This yields a configuration that has an equally or more clustered appearance than the standard MDS with the same transformation parameters. In the other variant (P-COPS), one can automatically select optimal transformation parameters so that the clustered appearance of the configuration is improved.

Formally, in COPS, one combines a normalized stress function $\sigma'(\mathbf{X}|\theta)$ given hyperparameter vector $\theta$ (for the transformations) and the OPTICS Cordillera ($OC'(\mathbf{X})$) to the following objective:

$$\sigma_{\text{COPS}}(\mathbf{X}|\theta) = v_1 \cdot \sigma'(\mathbf{X}|\theta) - v_2 \cdot OC'(\mathbf{X}), \tag{5}$$

with scalarization weights $v_1, v_2 \in \mathbb{R}_{\geq 0}$. The stress in Equation (5) is then minimized either over $\mathbf{X}$ (COPS-C) or over $\theta$ (P-COPS). Note we abused notation a bit here to convey the idea by only gave the target function for COPS-C; the one for P-COPS needs to be written a bit differently. We refer the interested reader to [1].

The scalarization weights $v_1, v_2$ are treated as given and in COPS-C are typically a convex combination $v_2 = 1 - v_1$ with $0 \leq v_1 \leq 1$. Minimizing $\sigma_{\text{COPS}}$ then jitters the configuration that would be obtained from minimizing $\sigma'$ alone towards a more clustered arrangement; the amount of jittering is governed by the values of $v_1, v_2$. For a given $\theta$, if $v_2 = 0$, the result of the above equation is the same as solving the respective MDS problem given by $\sigma'$. In general, we recommend to put much more weight on $v_1$ so as to stay faithful to the MDS result (say, $v_1 > 0.9$; the default in cops is $v_1 = 0.975$). It is also prudent to try out different starting configurations and choose the result that shows the overall lowest value for $\sigma_{\text{COPS}}(\mathbf{X}|\theta)$ [32].

In this article, we fit COPS-C with different given transformations for ratio, interval, s-stress and multiscale MDS. We will also use P-COPS to select clusteredness-optimal transformation parameters $\theta^* = (\kappa^*, \lambda^*)$ for p-stress and then run COPS-C with p-stress and the $\theta^*$ we found from P-COPS.

*2.5. Facets*

From a practical point of view, the important thing is the interpretation of a PS configuration. Obviously, one can interpret the pairwise distances between the points in the configuration plot. The closer two points, the less dissimilar the corresponding objects are, assuming the PS solution fits well. As opposed to principal component analysis (PCA), the interpretation of the $p$ dimensions is of less importance in PS [22], especially since the axes can be rotated and reflected arbitrarily without changing the stress value and therefore the actual fit.

Another avenue of interpretation is by facets. MDS has a long tradition in the field of facet theory [13,33] where researchers aim to partition a configuration space into connected, non-overlapping, and exhaustive parts, which are then subject to interpretation. A longstanding problem has been how to optimally identify these facets in a data-driven way. Ref. [14] propose to use support vector machines (SVM) [34,35] to solve this problem. The basic idea is to take a PS configuration and perform a supervised classification of the

configuration space with given facet labels, leading to a regional interpretation. For the supervised classification, SVM are used. SVM are non-probabilistic classifiers that allow for linear and nonlinear classification. A key component of any SVM is the kernel function, which allows mapping of the data into a transformed feature space where the classification problem becomes linear. In the original input space, this determines the functional shape of the classification boundaries used to separate the points into regions in the feature space (i.e., the PS configuration in our case). The most prominent kernels are the linear kernel (linear boundaries), the polynomial kernel (nonlinear boundaries using a polynomial of a particular degree), and the radial basis function (or Gaussian) kernel (nonlinear, radial boundaries; they depend only on the distance between the points and a center). We will use this approach in the data example below.

*2.6. Data Description*

The data for the present study are drawn from three randomized trials of the MATCH youth psychotherapy, each conducted in a community treatment setting by community clinicians working in these clinics [36–38]. All participants in these trials who received the MATCH youth psychotherapy (i.e., were not in a control condition) and who received at least one session of MATCH were included. The use of MATCH treatment modules was recorded by clinicians for each treatment session. Clinicians could use any MATCH treatment module in any order for each client and could use each module in as many sessions as they wished. The raw data are the counts of how many sessions each of the 32 MATCH-ADTC modules was used with each of the 449 patients over the course of their treatment. Thus, for each module, we have a count profile, for example, the module "GettingAcquainted" was used 2 times with the first patient, 1 time with the second, 0 times with the third and so on. We are interested in making statements about the usage of the modules, so the focus of our investigations are dissimilarities of the modules.

The profiles of the raw counts of how often each module has been used over the patients are what we use to calculate the matrix of pairwise $\phi$-distances $\delta_{ij} = d_{ij}^{\phi}$ between modules (function `phidistance` in package **cops**). Our distance matrix is available in **cops** via `data(matchphi)` containing the object `matchphi` (or directly as `cops::matchphi`). We create a new object `distM` to work with

```
R> distM <- cops::matchphi
```

which is visualized in Figure 1 (darker shades stand for less distance and lighter for more distance).

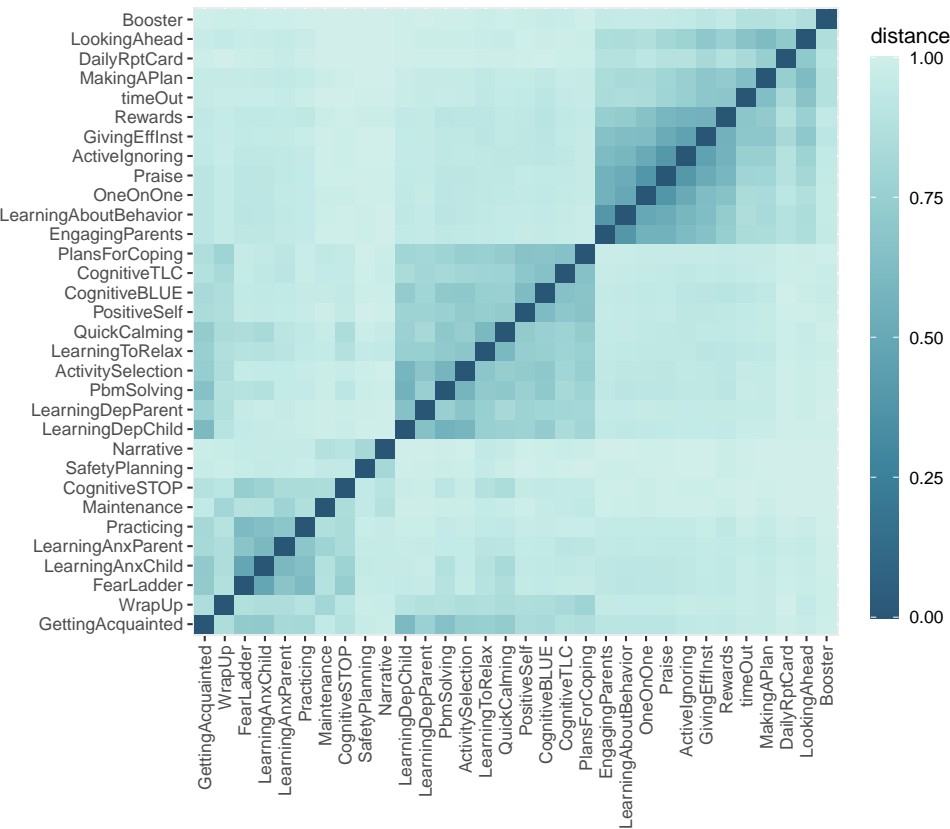

**Figure 1.** An image plot of the pairwise distance matrix with elements $\delta_{ij} = d_{ij}^{\phi}$ for the MATCH data. Darker shades stand for less distance.

We see that there seem to be blocks of modules that are closer together and more distant from other modules. We also see that the modules "GettingAcquainted" and "WrapUp", and to a lower degree "QuickCalming", "LearningToRelax" and "PbmSolving", are closer to more modules than what is typical for the data. "Booster", "Narrative" and "SafetyPlanning" seem to have the highest average distances to all other modules.

## 3. Results

In this section, we walk the reader through the analysis and results of the analysis strategy outlined above.

### 3.1. Clustering Structure in the Original Space

Since we are interested in exploring the similarity of the usage of MATCH-ADTC modules (via the count profile over the patients) and finding clusters based on module usage, we first start with looking at an OPTICS analysis of the data in the original space. For this, we use the implementation `optics` from package **dbscan** [39] and `cordillera` from **cordillera** [12].

We are especially interested in clusters that comprise at least three modules (that is the parameter $k$, the minimum number of points per cluster specified as `minPts = 3`). This was inspired partly by the FIRST principles ([7], see also Section 3.3), a theoretical category system for treatment techniques. Theoretically classifying the MATCH modules based on FIRST leads to categories that comprise at least three modules (with the exception of FeelingCalm comprising two). Together with the consideration that allowing for clusters with less than three modules is likely to generate a large number of clusters with 32 modules, leading to fragmentation, we chose $k = 3$. We do not expect any of the modules to be "just noise", so the $\epsilon$ parameter that defines the neighborhood around a point in which to look for other points just needs to be set sufficiently high to consider each point as a possible

neighbor of every other point; we use `eps=10` (10 times the maximum possible $\phi$-distance). Setting a smaller $\epsilon$ has the effect of letting points in sparsely populated regions be labeled as noise, which can make sense if we cannot take the data at face value because we suspect, e.g., contamination, error-prone measurement or similar in the data. For the $OC'(\mathbf{X})$ via `cordillera`, we use the same setup as for `optics` (so, `eps=10` and `minpts=3`) and measure it in terms of the $L^2$-norm (Euclidean), so `q=2`. As $d_{max}$, we will use the function default, which is the max $r_i^*$, making the $OC'(\mathbf{X})$ a goodness-of-clustering index relative to the largest reachability observed for the distance matrix.

We run OPTICS with this setup and also calculate the OPTICS Cordillera by

```
R> o1 <- dbscan::optics(as.dist(distM),minPts=3,eps=10)
R> c1 <- cordillera::cordillera(as.dist(distM),minpts=3,eps=10,q=2)
```

The $OC'(\mathbf{X})$ is around 0.14, which suggests that the modules are not very clustered based on the $\phi$-distance and `minpts=3`.

The corresponding reachability plot with the $OC(\mathbf{X})$ (black line) is shown in the top panel of Figure 2. We see there is some but not much up and down as captured by the $OC(\mathbf{X})$; mostly the reachabilities are relatively high, which means points are relatively far from their closest neighbors, and the valleys are not very deep meaning that the clusters existing in the original space are not very dense, with the two closest objects being modules "Praise" (module 24) and "OneOnOne" (23). They seem to belong to a cluster with modules "GivingEffInst" (26), "ActiveIgnoring" (25), "LearningAboutBehavior" (22) and "EngagingParents" (21). From visual inspection of the reachability plot, it looks as though there may be 4 or 5 clusters (valleys) in the original space, but these clusters are neither well-separated (average within cluster reachability is not much lower than the peaks) nor very cohesive (valleys are not deep). It also appears that the clusters are nested within bigger clusters (in Figure 2 to the left and right of the module "SafetyPlanning" (9)). Overall, the picture suggested in the original space is one of a relatively regular distribution of the modules, with little clustering information.

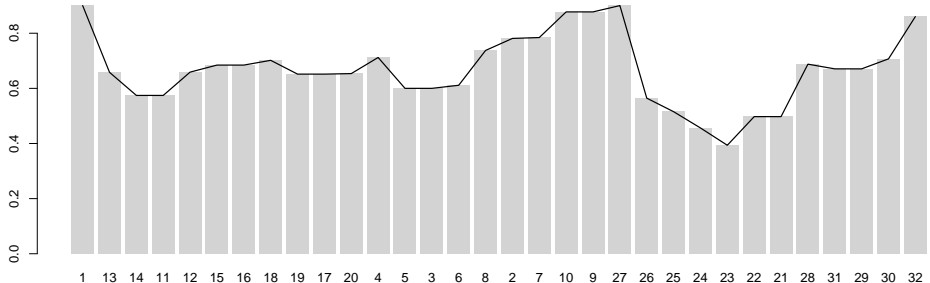

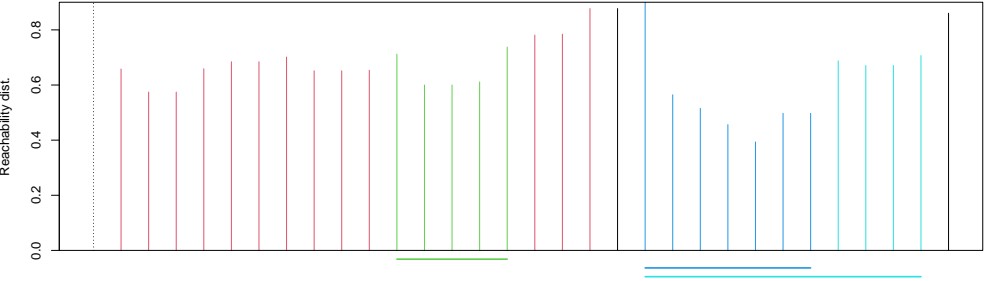

**Figure 2.** OPTICS reachability plot with stylized $OC(\mathbf{X})$ (top) and with coloring based on the clusters found by the $\xi$-strategy (bottom). In the lower panel, the cluster nesting is displayed below the plot.

The *OC*(**X**) already suggested that the clusteredness of the modules in the original space is not great. We will now use the *ζ*-strategy on our OPTICS result to suggest clusters (function `extractXi`); for this, we set the `xi` parameter to 0.039, which is just shy of the minimum reachability between modules 23 and 24. The `xi` parameter can be interpreted as detecting clusters in the reachability plot if there is a change of more than *ζ* in relative cluster density or in the steepness in the reachability plot. This may signal a new cluster. The value for `xi` needs to be specified by the user and often one has to try a few values to find one that is working well; in general the smaller it is, the more clusters are found.

```
R> cls <- dbscan::extractXi(o1,0.039)
```

The *ζ*-strategy with `xi=0.039` suggests that there are four clusters of at least 3 points with two modules not assigned (noise, the modules "Booster" (32) and "Safety Planning" (9)). The bottom panel in Figure 2 shows the reachabilities colored with the cluster assignment (black is noise) and the nested clusters as color lines in the lower part of the plot. There is a big cluster (red) containing a denser cluster (green); the latter are modules "LearningAnxChild" (4), "LearningAnxParent" (5), "FearLadder" (3), "Practicing" (6), and "CognitiveSTOP" (8) and they are nested within the larger red cluster additionally comprising modules "GettingAcquainted" (1), "PbmSolving" (13), "ActivitySelection" (14), "LearningDepChild" (11), "LearningDepParent" (12), "LearningToRelax" (15), "QuickCalming" (16), "CognitiveBLUE" (18) "CognitiveTLC" (19), "PositiveSelf" (17), "PlansForCoping" (20) (to the right of green cluster) and "WrapUp" (2), "Maintenance" (7), and "Narrative" (10) (to the left of green). The blue cluster comprises "Rewards" (27), 26, 25, 24, 23, 22, 21 and is nested in the turquoise one that additionally comprises "timeOut" (28), "LookingAhead" (31), "MakingAPlan" (29) and "DailyRptCard" (30) on lower density.

So far, we have learned the following: In the original space, clusteredness is not high and the clustering is not particularly suggestive, with two large clusters containing two nested clusters each. The large cluster (red) contains roughly two-thirds of the modules, those that are anxiety-, trauma- and depression-related, as well as the modules that are used for starting and ending a treatment. Within that cluster, "LearningAnxParent", "LearningAnxChild", "FearLadder", "Practicing" and "CognitiveSTOP" form a denser cluster. The other cluster (turquoise) comprises roughly the other one-third of the modules, which are the conduct-based modules with the dense cluster around "OneOnOne" and "Praise".

*3.2. Scaling with COPS*

While the OPTICS analysis leaves us wanting a bit, we now turn to scaling the modules in a reduced space. This allows us not only to visualize the spatial arrangement of the modules in a Cartesian space, but perhaps suggests clusterings that we have not been able to detect before.

With the R package **cops**, we can fit two instances of the COPS framework: First, COPS-C with the fitting function `copstressMin` or using `cops` and specifying `variant = 'COPS-C'`. Second, we can also fit P-COPS (returning optimal parameter for p-stress), either with the fitting function `pcops` or `cops` with `variant='P-COPS'`. In this article, we will focus on the COPS-C functionality.

Each variant's fitting function has a number of arguments to allow for many different types of models to be fitted. We give a short overview of the main ones in this article; a more comprehensive discussion is available in the package vignette to **cops** (obtainable in R via `vignette('cops')`).

In `copstressMin`, we can choose the optimal scaling transformation with the argument `type` to be `'ratio'`, `'interval'` or `'ordinal'`. If a p-stress COPS (or any of its special cases) should be fitted, we can set the `kappa`, `lambda` and/or `nu` parameters to the desired values in `copstressMin`; note then only ratio transformations are allowed, so `type = 'ratio'` only. In what follows, we will use COPS with the following stresses: First, we look at a ratio COPS-C with $v_2 = 0$ (corresponding to standard ratio MDS as reference). We then fit a ratio COPS-C (`type = 'ratio'`), interval COPS-C (`type = 'interval'`) and

two specific p-stress models, multiscale COPS-C (with `distM2 = log(1 + distM)`, `type = 'ratio'` and `kappa = 0.1`) and s-stress COPS-C (`type = 'ratio'`, `kappa = 2`, `lambda = 2`). We also run a P-COPS with `loss = 'powermds'` (i.e., p-stress without weights) and select an optimal $\kappa, \lambda$ for p-stress. We then refit a p-stress COPS-C (`type = 'ratio'`) with the optimal $\kappa^*$ and $\lambda^*$ as respective arguments for `kappa` and `lambda`.

For COPS models, we can set the weight $v_1$ of the stress measure with `stressweight` and the weight $v_2$ for the $OC'(\mathbf{X})$ with `cordweight`. We use $v_1 = 0.975$ (`stressweight = 0.975`) and $v_2 = 0.025$ (`cordweight = 0.025`) in all subsequent COPS-C models apart from standard ratio MDS, so essentially we are willing to sacrifice up to 2.5% of fit for a more clustered appearance. This is a fairly low value (and the default), so this will lead to a configuration that is still very close to the optimal one for the corresponding standard MDS (which can be obtained by setting $v_2 = 0$). The choice of $v_1$ and $v_2$ is up to the user, but we recommend to use relatively high values of $v_1$ (e.g., $v_1 > 0.95$ and $v_2 < 0.05$). We also point out that in terms of optimization, a $v_2$ that is close to 0 makes the complicated optimization of COPS problem less difficult. The weighting for P-COPS works a bit differently, but there is a smart default (see [1] for details).

The number of dimensions of the target space can be changed with `ndim`. It is also possible to specify a symmetric weight matrix `weightmat` and an initial configuration `init`, which can be used to try out different starting configurations. The minimization of (5) is complicated and there are many different heuristics implemented; the optimization strategy can be changed with the argument `optimmethod`. The default strategy usually works well, so we stick to that one. We let the solver in COPS-C use up to `itmax=50000` iterations, which will likely allow convergence of the optimizer at the default convergence criterion. We note that such a high `itmax` is used for illustration here and is overkill in most applications; normally we use between 5000 and 10,000.

What is left are the arguments for measuring clusteredness with the $OC'(\mathbf{X})$. These are the same ones as in `cordillera` and mainly are `minpts` (minimum number of points $k$ to make up a cluster) and `dmax` the maximum reference distance and winsorization limit. The decision on $k$ must be made on substantive grounds; we will use `minpts = 3`. As `dmax`, we set 1, which corresponds to around 1.5 times the maximum reachability in the standard ratio MDS; this provides robustness to the $OC'(\mathbf{X})$ against noise points or outliers that have larger reachabilities than that. We may also set `q`, which we keep at 2 and `epsilon`, but that can just be set to some high number (in our case the default, which is 10). Note we now want to compare different $\mathbf{X}$, so we need to have one $d_{max}$ that defines the maximal clusteredness for all; then, the $OC'(\mathbf{X})$ is the clusteredness achieved relative to that maximum for all $\mathbf{X}$. When looking at only one $\mathbf{X}$, one can just use the default for goodness-of-clustering, but that default is likely to change between different $\mathbf{X}$. So, when comparing different $\mathbf{X}$ we should use the same $d_{max}, k, q, \epsilon$.

So, let us set up the control parameters in R.

```
R> ## OC parameters
R> minpts <- 3
R> dmax <- 1
R> q <- 2
R> ##iterations
R> itm <- 50000
```

Our starting point is the standard ratio MDS solution, which we can obtain by setting `stressweight = 1` and `cordweight = 0` in the call to `copstressMin`.

```
R> ## Standard MDS
R> MDSM0 <- cops::copstressMin(distM,            #distance object
type="ratio",                 #MDS type
dmax=dmax,minpts=minpts,q=q,  #metaparameters
stressweight=1,cordweight=0,  #v1 and v2
itmax=itm)
```

This solution has a stress-1 (square root of stress) of 0.315 and an $OC'(\mathbf{X}) = 0.11$ for the respective Cordillera parameters. As before with our OPTICS investigation in the original space, in terms of clusteredness this is relatively low.

The configuration plot and the corresponding reachability plot with stylized $OC(\mathbf{X})$ can be found in Figure 3. They corroborate that the clusteredness obtained for this result is not very high and that the arrangement of modules has a relatively low density for all. Nevertheless, we can already see which modules seem to be used similarly over the patients: In the top left quadrant, we find modules that are related to depression, whereas in the bottom left quadrant we find modules related to anxiety. In the top right and bottom right quadrant, we have modules that are related to conduct problems. The trauma related modules are located between the anxiety modules and the conduct modules. The icebreaker module ("GettingAcquainted") and the finisher ("WrapUp") are located between the depression and anxiety modules. Still, the modules are not arranged in a way that allows us to identify a more detailed organizing principle.

To change that, we now put some more weight on the Cordillera as described above by setting `stressweight=sw` and `cordweight=cw`.

```
R> ## Weights
R> cw <- 0.025
R> sw <- 0.975
```

The change in clusteredness is quite drastic: Stress-1 became only slightly worse (0.319), but the $OC'(\mathbf{X})$ more than quadrupled to $OC'(\mathbf{X}) = 0.47$. This is also easy to discern in the configuration plot and the corresponding reachability plot, which can be found in Figure 4. The reachability plot is much more rugged now, reflecting the higher clusteredness and clearer clustering structure as seen in the configuration plot.

This result opens up many new avenues for interpretation: The basic quadrant structure we already identified in the standard MDS is upheld. Within those, however, we now find new and clearer clusters of modules. For example, it seems that "GettingAcquainted" has a similar profile to the relaxation modules, suggesting that they are used similarly over the youths. The anxiety module "FearLadder" and the learning about anxiety modules now cluster more strongly and "Practicing" is moved closer to them. The trauma modules are still separated clearly from the anxiety modules, but are close. We find additional clusters also for the conduct modules, which seem to group in three clusters: the clusters of "EngagingParents", "LearningAboutBehavior" and "OneOnOne", a cluster of "GivingEffInst", "Praise", "Rewards", "ActiveIgnoring" and one comprising "timeOut", "MakingAPlan", "LookingAhead" and "DailyRptCard". "Booster" appears removed from any clusters. Among the depression modules, the modules are separated into two more obvious clusters, each with a denser sub-cluster: the aforementioned "GettingAcquainted" and relaxation module cluster on the one hand and "LearningDepChild","LearningDepParent" and "ActivitySelection" on the other. There is already much to be hypothesized from this result, but we will look at whether there are any changes if we use other stress models as well.

We next use COPS-C with the interval transformation. Compared to ratio MDS, this essentially removes shared absolute information in the proximities from the fitted distances in the configuration due to allowing a constant in the disparities. A fitted distance of 0 then represents the shared absolute information and the fitted distances only approximate the information that is left, therefore emphasizing the dissimilarity difference more strongly; this increases the coefficient of variation of the $\hat{d}_{ij}(\mathbf{X})$ and we expect a clearer clustering structure.

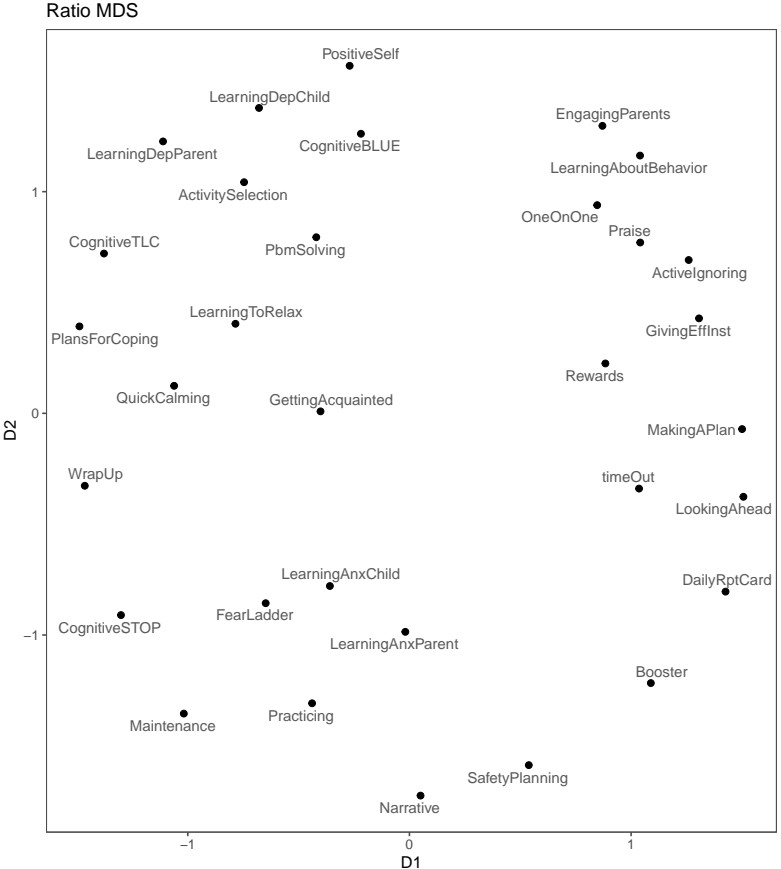

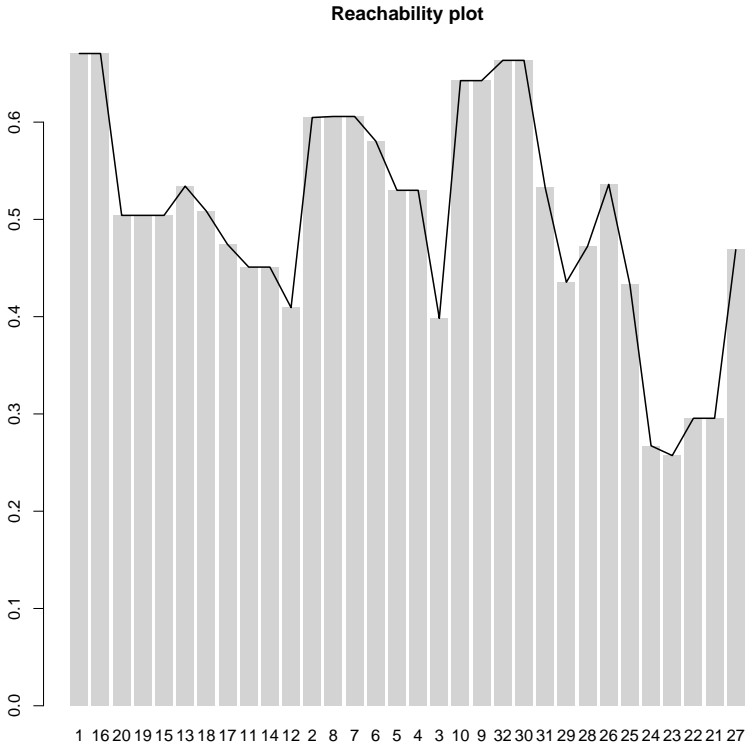

**Figure 3.** Configuration plot (top) and reachability plot (bottom) with stylized OPTICS Cordillera (black line) for the standard ratio MDS of the MATCH-ADTC modules.

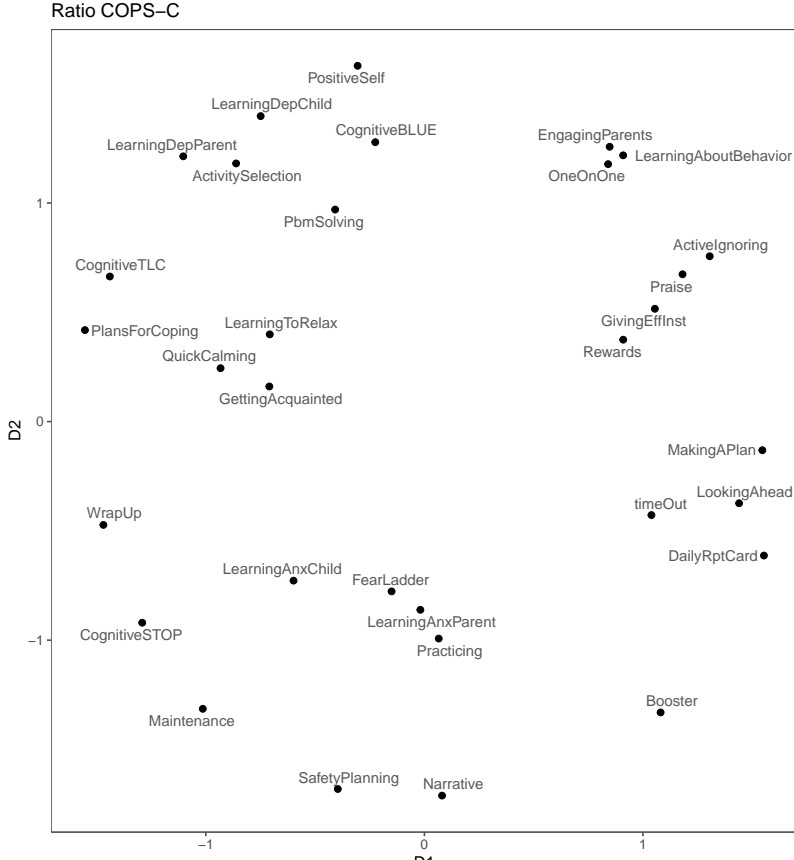

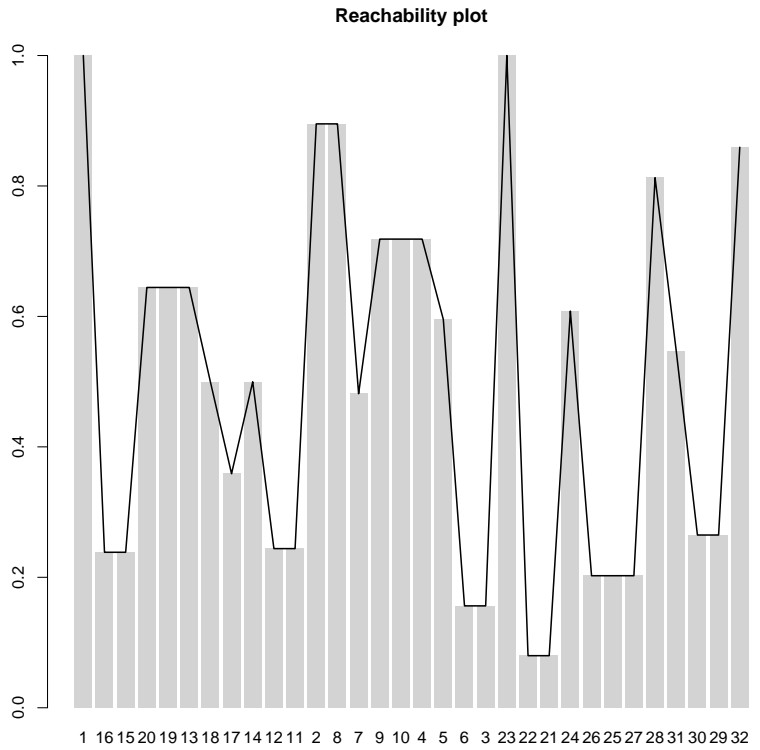

**Figure 4.** Configuration plot (top) and reachability plot (bottom) with stylized OPTICS Cordillera (black line) for the ratio COPS-C of the MATCH-ADTC modules.

```
R> ## Interval COPS-C
R> MDSMint <- cops::copstressMin(distM, type="interval",
dmax=dmax, minpts=minpts, q=q,
stressweight=sw, cordweight=cw,
itmax=itm)
```

This solution has a stress-1 of 0.22 and $OC'(\mathbf{X}) = 0.49$. The configuration plot and the corresponding reachability plot can be found in Figure 5. Notice the winsorization effect of $d_{max}$ for the reachabilities for modules $9, 31, 32$.

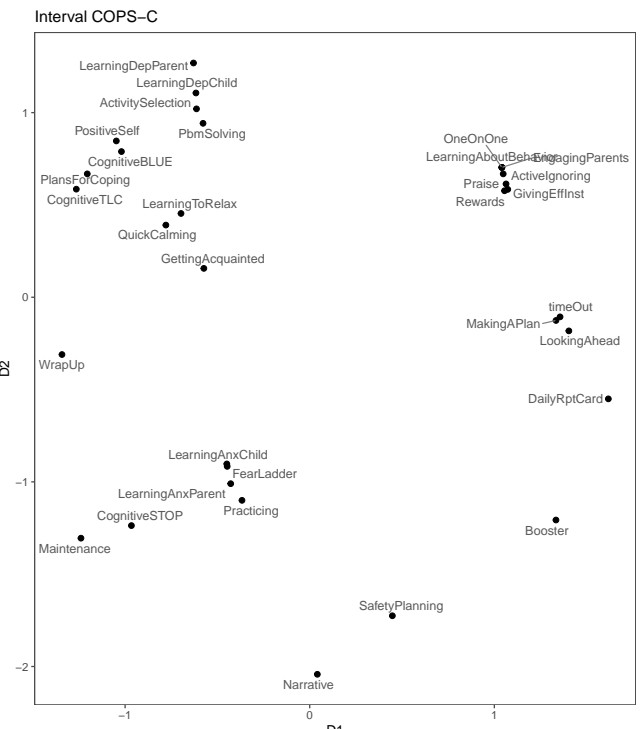

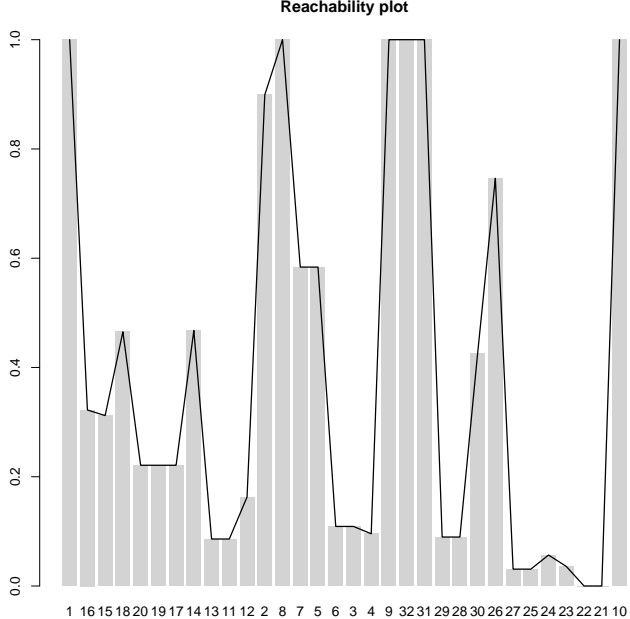

**Figure 5.** Configuration plot (top) and reachability plot (bottom) with stylized OPTICS Cordillera (black line) for interval COPS-C of the MATCH-ADTC modules.

The clustering has indeed become clearer. In particular, clusters are now much denser than they were in the ratio solution owing to the interval transformation. We also see that within the overall organizing principle we identified before, clusters of modules are emphasized or new ones are revealed. For example, "Quick Calming" and "Learning-ToRelax" are now still close to "GettingAcquainted" but form their own tuple. They are also located in between depression and anxiety modules. The conduct modules "En-gagingParents", "LearningAboutBehavior", "OneOnOne", "Praise", "ActiveIgnoring", "Rewards" and "GivingEffInst" form a dense cluster and so do "MakingAPlan", "timeOut", and "LookingAhead". The two trauma modules are now more clearly separated from the anxiety modules, but still relatively close. The anxiety modules form a bigger cluster with a denser nested cluster of "FearLadder", "LearnAnxParent", "LearnAnxChild" and "Practicing". The non-relaxation related depression modules can be separated into two groups: "LearningDepChild", "LearningDepParent", "ActivitySelection", "PbmSolving" on the one hand and "PositiveSelf", "CognitiveBLUE", "CognitiveTLC", "PlansForCoping" on the other.

We now turn to see if there is additional insight we can gain from fitting other stresses. We first fit an s-stress COPS-C model, which emphasizes larger distances more strongly as ratio or interval COPS-C did; this may allow us to learn something new by focusing mostly on the modules with the largest profile distances. We then fit a multiscale COPS-C that does basically the opposite of s-stress by emphasizing local distances and variation more strongly because smaller distances are made relatively larger and larger distances are made relatively smaller by the log transformation. The respective configuration plots and the corresponding reachability plots can be found in Figures 6 and 7.

We start with s-stress.

```
R> ## S-Stress COPS-C
R> MDSMsst <- cops::copstressMin(distM, type="ratio", kappa=2, lambda=2,
dmax=dmax, minpts=minpts, q=q,
stressweight=sw, cordweight=cw,
itmax=itm)
```

The s-stress COPS-C gives a stress-1 of 0.419 and an $OC'(\mathbf{X}) = 0.52$. In Figure 6, we see that clusteredness is high for the s-stress model (0.52), leading to many dense clusters of around 3 modules, which are often also nested within a larger cluster. The general story that we discussed in ratio and interval COPS-C is largely reproduced here, but emphasized: The conduct and depression modules cluster together and are organized in a number of similar clusters to what we had before, but the clusters are much denser. What is new is that with s-stress the trauma and anxiety modules now form one cluster with two very dense nested ones. "Booster" and "WrapUp" are individually singled out as being used differently to all other modules.

For multiscale COPS-C, we manually transform the $\delta_{ij}$ to $\log(1 + \delta_{ij})$ because they lie between 0 and 1, and we need non-negative dissimilarities. We further use the p-stress approximation to multiscale stress with $\kappa = 0.1$.

```
R> distM2 <- log(1+distM)
R> diag(distM2) <- 0
R> MDSMmsc <- cops::copstressMin(distM2, type="ratio", kappa=0.1,
dmax=dmax, minpts=minpts, q=q,
stressweight=sw, cordweight=cw,
itmax=itm)
```

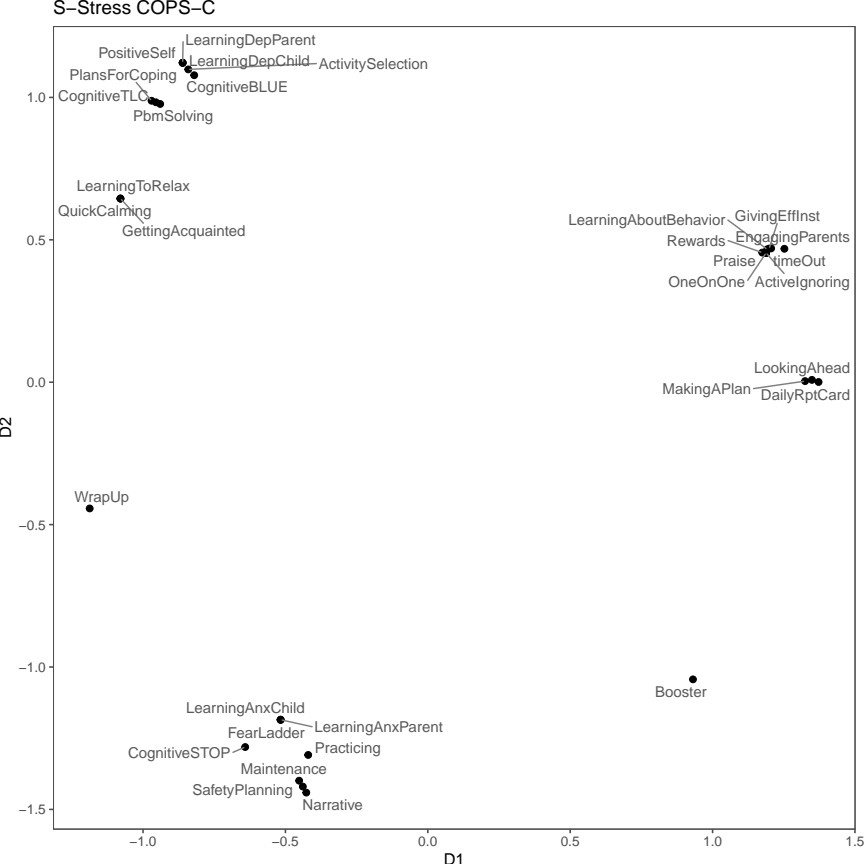

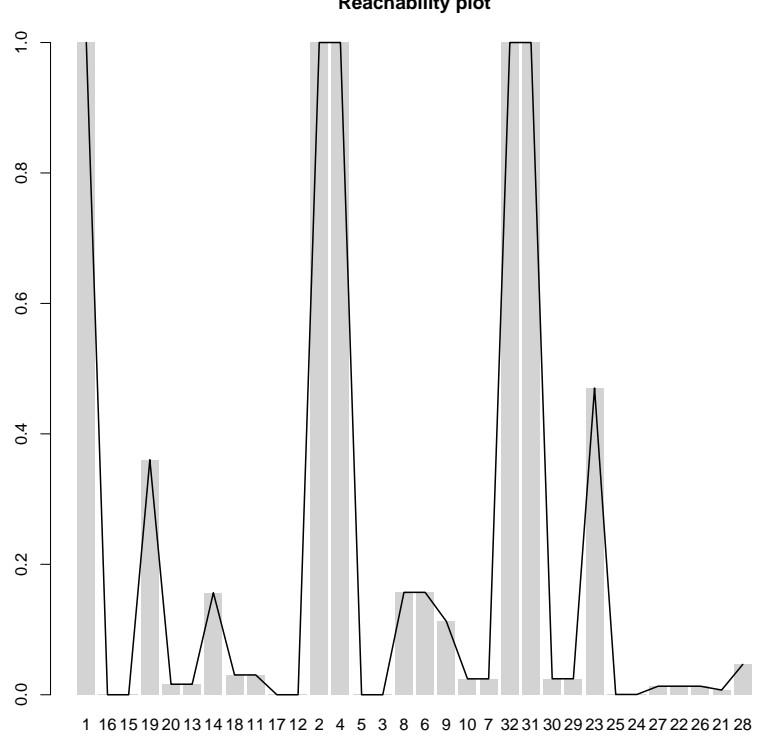

**Figure 6.** Configuration plot (top) and reachability plot (bottom) with stylized OPTICS Cordillera (black line) for s-stress COPS-C of the MATCH-ADTC modules.

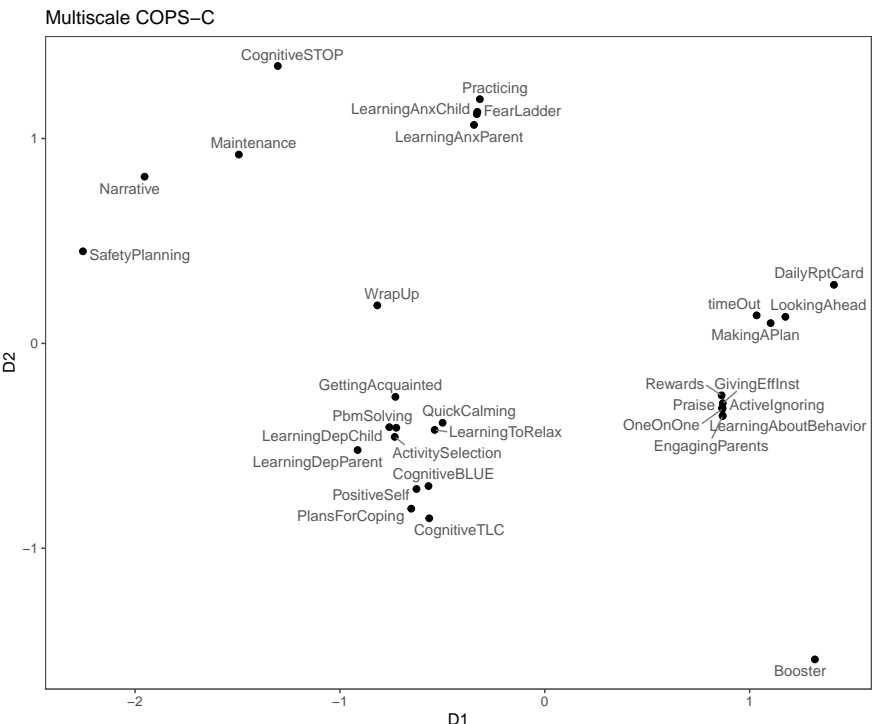

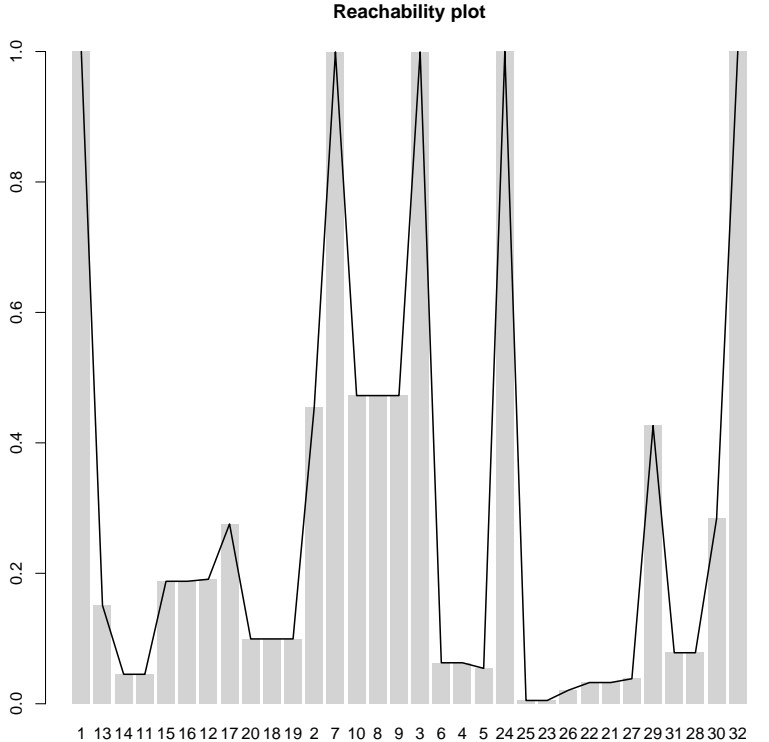

**Figure 7.** Configuration plot (top) and reachability plot (bottom) with stylized OPTICS Cordillera (black line) for multiscale COPS-C of the MATCH-ADTC modules.

For multiscale COPS-C (see Figure 7), we again have high clusteredness (0.51). While confirming some of the already established patterns, we also find new things: Of note is that "WrapUp" takes the center stage and is less close to "GettingAcquainted" as in the ratio and interval. The anxiety modules seem to fall into one cluster apart from "CognitiveSTOP" and "Maintenance", the latter now bridging the anxiety and trauma modules. "Booster" seems to be used very differently to all the other modules, even the related conduct modules. To a certain degree, this was already suggested by prior analyses, but only in multiscale is "Booster" heavily represented as an outlier, mirroring the results from the OPTICS analysis, where it was considered to be a noise point.

Ratio, s-stress, and multiscale MDS are all special cases of p-stress resulting from different exponents of the power transformations $\kappa$ and $\lambda$. The COPS framework also allows us to use this to search for optimal power transformations to obtain an MDS model that is highly clustered, without having to penalize the fit itself when looking for the configuration (as in COPS-C). This method is called P-COPS and is a special case of a larger framework, the STOPS framework, which allows us to select parameters in MDS based on structural considerations [2]. We can run P-COPS for our data and see which $\theta^* = (\kappa^*, \lambda^*)^\top$ is suggested. We use a p-stress without weights (`loss='powermds'`) as the badness-of-fit measure. We specify to search for an optimal $\kappa, \lambda$ in the region $[0.01, 5]$ for each parameter (thus encompassing ratio, s-stress and multiscale). This is done with the arguments `lower` and `upper` that take a numeric vector of the same length as $\theta$ (here 2) with the lower and upper boundaries for the parameters, respectively.

```
R> ## P–COPS (powermds loss)
R> MDSMpco <- cops::pcops(distM, loss="powermds",
minpts=minpts, rang=c(0,dmax),
upper=c(5,5), lower=c(0.01,0.01),
itmaxi=10000)
```

The optimal $\theta^*$ found is $(0.467, 3.644)^\top$, which are in `MDSMpco$par`. Note that this means that the ratio, multiscale or s-stress models were not selected as optimal; rather the selected MDS model is close to $\hat{\delta}_{ij} = |b|\delta_{ij}^{3.6}$ and $\hat{d}_{ij}(X) = \sqrt{d_{ij}(X)}$, which means it puts a lot of emphasis on the large dissimilarities and maps them to a concave function of the Euclidean distances emphasizing the variation in the smaller distances. The $OC'(X)$ for this MDS model is 0.37, which is quite high given that we only achieved it by changing the transformation parameters and not by penalizing the configuration.

Nevertheless, we can combine the two: We can use the parameters found with P-COPS and fit a p-stress COPS-C with said $\kappa^*$ and $\lambda^*$ to further improve the clusteredness of that configuration. We also use the P-COPS configuration as the starting configuration `init`.

```
R> ## Power Stress COPS–C
R> MDSMpst <- cops::copstressMin(distM,
kappa=MDSMpco$par[1], lambda=MDSMpco$par[2],
init=MDSMpco$fit$conf, dmax=dmax, minpts=minpts, q=q,
stressweight=sw, cordweight=cw,
itmax=itm)
```

The stress-1 for this p-stress COPS-C is 0.091 and the $OC'(X) = 0.43$. The resulting configuration plot and the corresponding reachability plots can be found in Figure 8.

We see that this configuration seems like a compromise of the results of the ratio, multiscale and s-stress models: dense and well separated clusters, "GettingAcquainted" and "WrapUp" are again scaled closer together, the conduct, depression, anxiety and trauma modules are grouped in well-separated clusters and within these clusters the scaling seems to suggest the nested subclusters already mentioned. "Booster" is once again clearly singled out as an outlier.

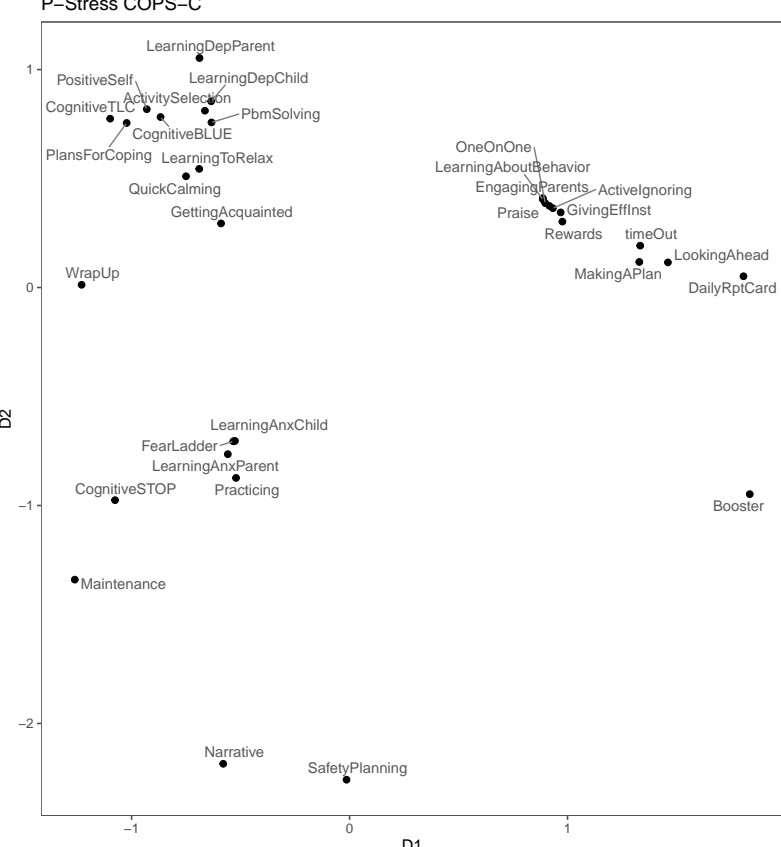

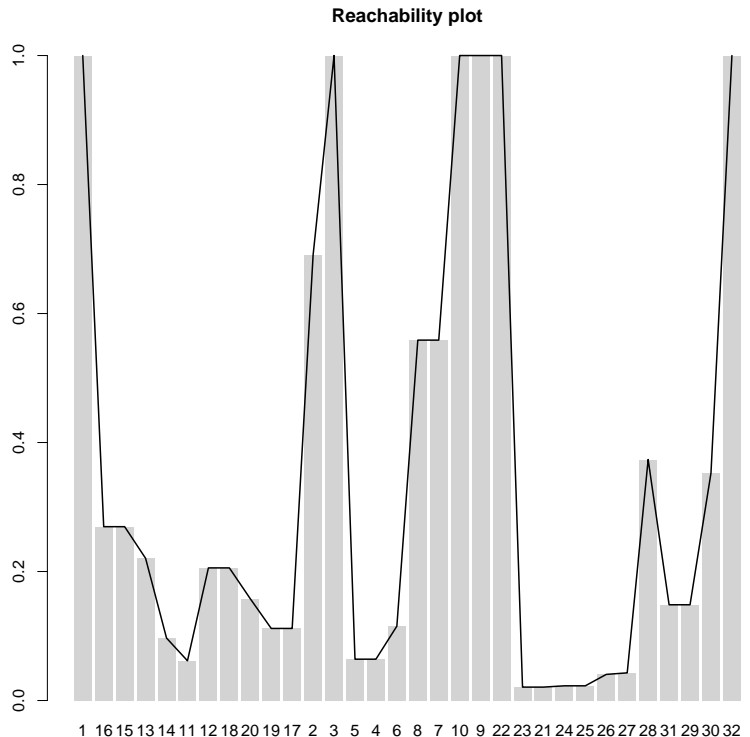

**Figure 8.** Configuration plot (top) and reachability plot (bottom) with stylized OPTICS Cordillera (black line) for the standard ratio MDS of the MATCH-ADTC modules.

### 3.3. Faceting with COPS and SVM

We already alluded numerous times to a congruence between the spatial arrangement of the modules in the COPS results and the behavioral health concerns that MATCH-ADTC was developed for. Note that we arrived on that in an exploratory fashion; in this section, we formally analyze this congruence with the facet framework of [14].

We use SVM to identify facets in the reduced space corresponding to the problems that MATCH addresses (called protocols): anxiety (ANX), depression (DEP), trauma (TRA) and conduct (CON). We also define the protocol General (GEN) to capture the cross-protocol modules "GettingAcquainted" and "WrapUp".

We will use the results of the interval COPS-C and of the p-stress COPS-C to derive our facets. We fit SVM with a radial kernel. We tune the SVM over costs of a constraint violation of 10 to 100 in steps of 2. The optimal cost parameter of the SVM is selected via a 10-fold cross-validation. For the best SVMs (optimal cost parameters of 16 and 18 for interval and p-stress COPS, respectively) we obtain perfect predictions: The partitioning of the COPS space allows us to perfectly allocate each module to its respective protocol. This is displayed in Figure 9, where we used `svm_mdsplot` from **smacof**, which also works for objects returned by `cops`. The most important information in this plot are the colored regions. To read the labels that are overplotted, one can refer to the configuration of points in Figures 4 and 8, respectively, as they are the same ones.

Each protocol has a region associated with it in the respective COPS-C space, and each module's assignment to its corresponding facet is perfect. There is also a circumplex-type structure of the protocol facets, meaning that trauma is located next to anxiety, anxiety next to depression and depression next to conduct. Conduct then also borders on trauma, although the modules of each protocol are further apart (especially in the p-stress result).

When exploring the configurations with the facets in Figure 9, we see that the scaling results suggest clusters within each facet. To explore this further, we made use of the FIRST principles [7] that provide a theory-based categorization of the modules into empirically supported principles of change (ESPCs), or common cross-cutting treatment techniques in evidence-based youth psychotherapies. These ESPCs were derived via a review of relevant literature, e.g., [40,41], and subsequently by a panel of experts consisting of both researchers and practitioners [7]. These ESPCs include "Relationship Building" (e.g., get-to-know-you activities and psycho-education; RelBuild), "Future Planning" (e.g., discussing how a skill might be used in the future and how to identify relapses; FutPlan), "Trying the Opposite" (e.g., exposure to feared stimuli and behavioral activation; TTO), "Repairing Thoughts" (e.g., working to change hostile attributions and excessively negative thinking patterns; RepThou), "Solve Problems" (e.g., naming the problem and thinking of possible solutions; SolvProb), "Increasing Motivation" (e.g., rewards for desired behaviors or time outs; IncMot) and "Feeling Calm" (e.g., deep, slow breathing and calming imagery; FeelCalm). Each module belongs to one ESPC. We thus classify the modules based on protocol and ESPC into a combined category. For example, "Practicing" is ANX-TTO and "Narrative" is TRA-TTO. We are interested in two aspects: a) within each protocol are there modules that are used similarly and are of the same ESPC or b) within each protocol, do the similar modules vary with respect to ESPCs, so do clinicians use modules from different EPSC similarly?

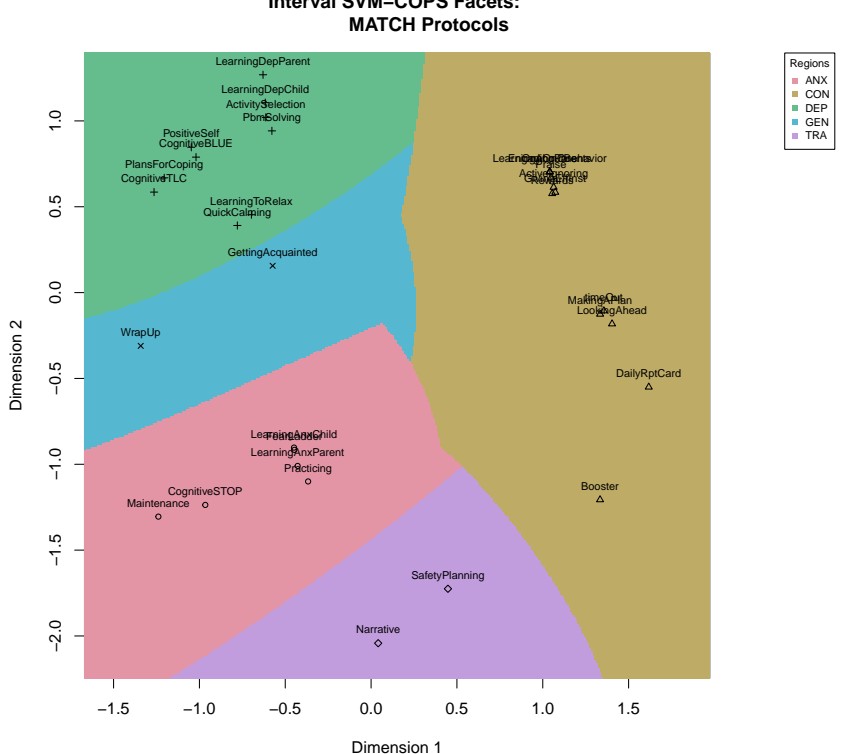

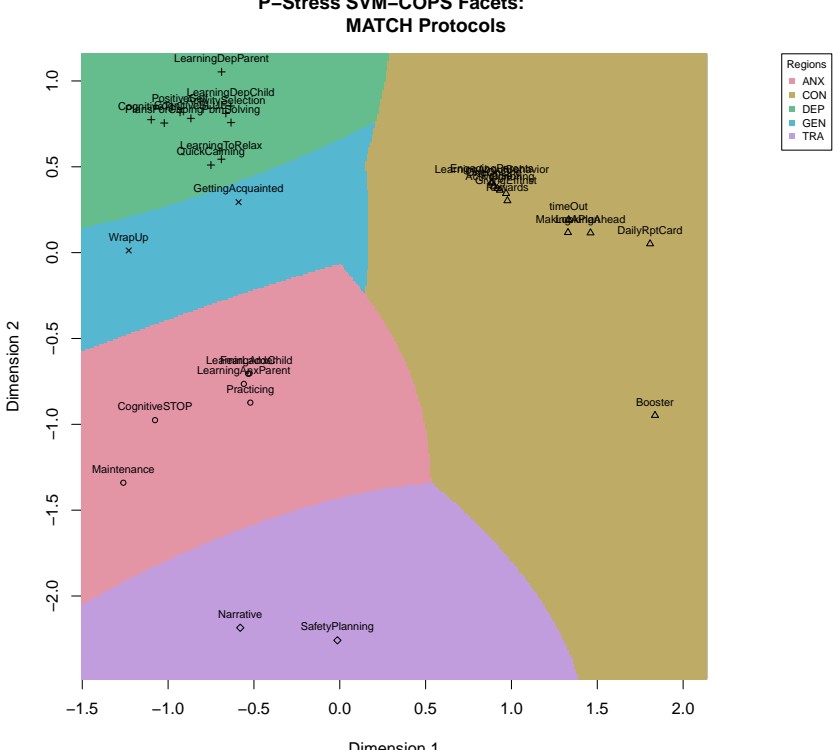

**Figure 9.** SVM-MDS facet plot for the interval COPS-C solution (top) and the p-stress COPS-C (bottom) based on the MATCH protocols anxiety (ANX, red), depression (DEP, green), conduct (CON, brown) and trauma (TRA, pink), as well as general (GEN, blue).

To that end, we use the results from ratio and interval COPS-C and color the modules based on their combined Protocol-ESPC categorization. This is shown in Figure 10, where in both results we see that in the top left quadrant, the depression facet, modules of different ESPC cluster together (e.g., for interval COPS-C "LearningDepChild" and "LearningDepParent", which are RelBuild with "ActivitySelection", which is TTO and "PbmSolving", which is SolveProb). The use of the modules in this facet appears relatively diverse; if we look at the ratio COPS-C result then the depression facet shows a fairly equidistant use of modules from six different ESPCs (RelBuild, TTO, RepThou, FutPlan, FeelCalm and SolvProb). In the interval COPS-C there is fragmentation into more homogeneous subclusters, but there are still at least three ESPCs per cluster (out of four modules). The one exception are the FeelCalm modules, which are of the same ESPC.

For anxiety, the picture is a bit different. In both COPS-C results, there is a cluster with three RelBuild modules ("LearningAnxChild", "LearningAnxParent", "FearLadder") and also "Practicing" (TTO). "Maintenance" and "CognitiveSTOP" are a bit removed and form a relatively dispersed cluster with the RelBuild ones. Overall, in the anxiety facet, the use of the modules is less diverse than it was with depression. This is partially attributable to the anxiety facet only comprising modules of four ESPCs, but still the anxiety modules show a more homogeneous picture due to the ANX-RelBuild modules scaled close together.

For conduct modules, we find two dense clusters of at least three modules (interval COPS-C) or three clusters of at least three modules (ratio COPS-C) with varying densities. In the ratio COPS-C, we see that two out of three clusters are relatively homogeneous with respect to the ESPCs. The cluster of "ActiveIgnoring", "Praise", "Rewards" and "GivingEffInst" is perfectly homogeneous with instances of the IncMot ESPC. The other dense cluster contains "LearningAboutBehavior" and "EngagingParents", which are both RelBuild and they cluster with "OneOnOne", which is IncMot. In the interval COPS-C, these two clusters are merged into one very dense cluster, making this a relatively homogeneous cluster comprising seven modules of only two ESPCs (IncMot and RelBuild). There is also a relatively diverse dense cluster in the conduct facet, which is the one comprising "MakingAPlan" (SolvProb), "timeOut" (IncMot), "LookingAhead" (FutPlan) in interval COPS-C, which presents as a dispersed cluster enhanced with "DailyRptCard" (IncMot) in the ratio COPS-C.

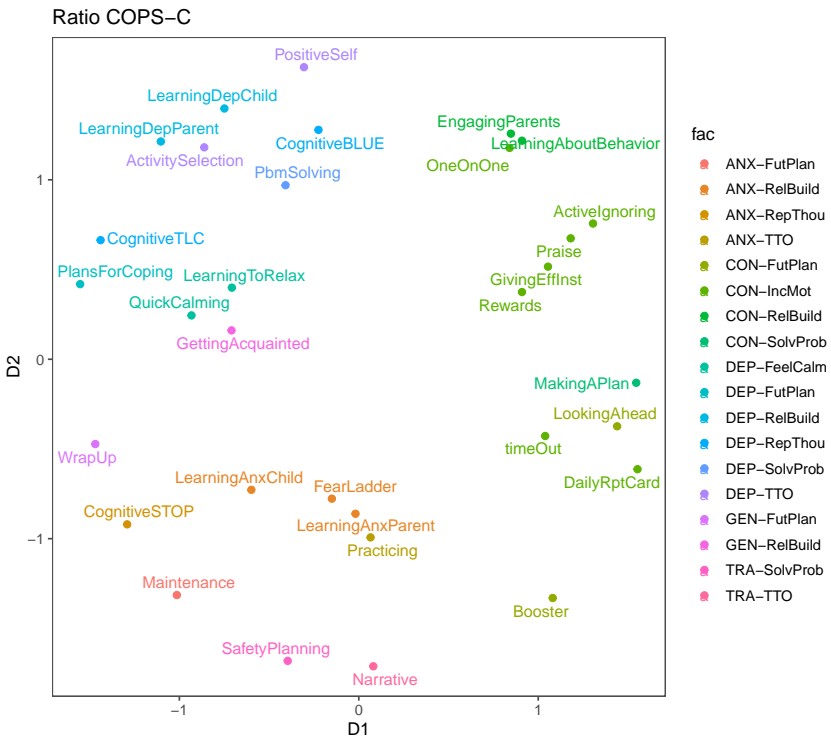

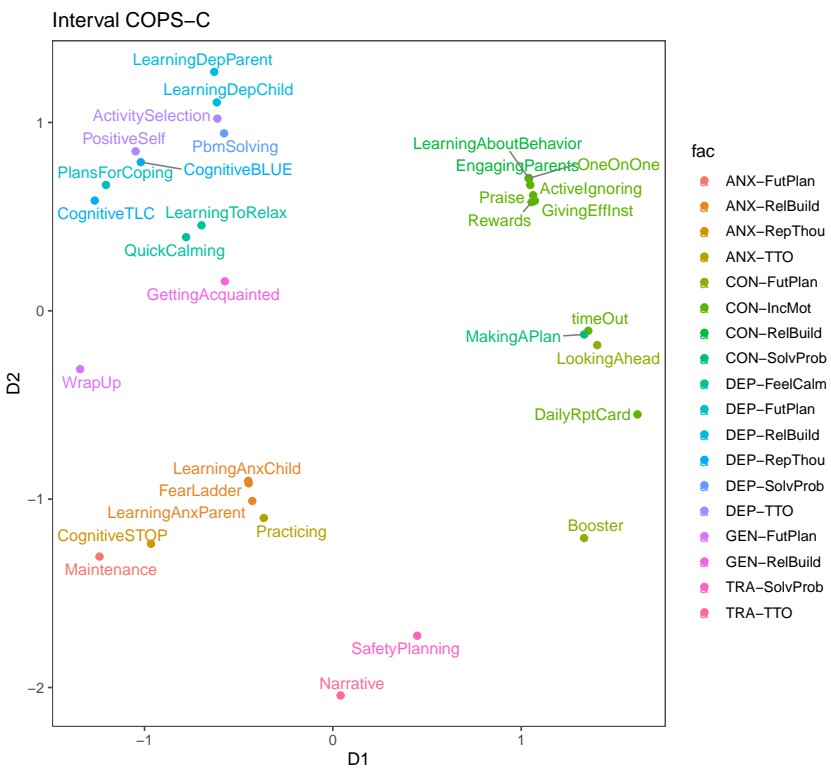

**Figure 10.** Configuration plot for the interval COPS-C solution (top) and the p-stress COPS-C (bottom) based on the combined MATCH protocols and ESPCs categorization.

## 4. Discussion

In this article, we illustrated a modern exploratory data analysis approach centered around a novel variant of multidimensional scaling, Cluster Optimized Proximity Scaling (COPS), which we used to explore data on clinicians' usage of the MATCH-ADTC for behavioral health problems in youths. COPS combines badness-of-fit measures used in MDS with an index for clusteredness of the configuration, the OPTICS Cordillera. It provides a scaling that emphasizes clusteredness in the configuration. To complement the scaling approach, we also used and illustrated the OPTICS clustering algorithm in the original space. COPS and OPTICS made use of pairwise $\phi$-distances between the modules based on the profile of how often each module was used over the patients; in the $\phi$-distance, modules with similar count data profiles are closer. We toured through a few stress functions and data transformations to highlight different aspects in the data, including COPS models employing ratio and interval transformations in Kruskal's stress and multiscale stress, s-stress and p-stress. In doing so, we also illustrated the usage of the R package **cops**, a comprehensive software suite for flexible multidimensional scaling in the COPS framework.

When looking at our data example, we found that COPS provided a scaling of the MATCH-ADTC that is sensible and allows us to visually infer clusters of modules from the configuration. The COPS analyses highlighted many clusters of modules that open up possible new interpretations of how MATCH-ADTC is used. For example, the icebreaker "GettingAcquainted" and the relaxation modules ("LearningToRelax" and "QuickCalming") are often scaled close to each other, sometimes together with the finisher "WrapUp". Additionally, "ActiveIgnoring", "Praise", "Rewards", "GivingEffInst" cluster together throughout the analyses. "FearLadder", "LearningAnxChild", "LearningAnxParent" and "Practicing" are used similarly over the youths and show up as a cluster. "SafetyPlanning" and "Narrative" form a recurring cluster as well.

In general, the modules were scaled much in accordance with the MATCH protocol they belong to (anxiety, depression, trauma, conduct or general). This was corroborated by a facet analysis with SVM-MDS to define regions corresponding to the protocols in the COPS space; we observed the perfect classification of modules to protocol facets. In hindsight, the obtained scaling, the clusters and the facets are often not surprising when we look at how clinicians use the MATCH process. They start with an initial primary problem protocol and then are guided through a standard course of modules along a flowchart. If new problems arise, then the instruction guides the clinicians into choosing another protocol and following a standard course of modules to help with that. Therefore, finding the modules grouped by the protocols and also within the facets to be grouped along the standard course of modules is sensible. The COPS analysis of the usage of MATCH-ADTC in many respects reflects the MATCH-ADTC flowcharts and instruction.

Of note is that while the protocols are recovered by COPS, they are not completely recovered by the OPTICS clustering in the original space. What is interesting is that after scaling and faceting the OPTICS clustering is discernible in the COPS results: If we take the decision boundary of the conduct modules as splitting our space, we have almost perfect correspondence of the two large OPTICS clusters to the conduct facet and a meta facet comprising the rest in the COPS results. Thus, the COPS results not only suggest a faceting that matches the clusters suggested by OPTICS's $\xi$-strategy in the original space, but also allow to differentiate more nested clusters, organize modules, protocols and clusters more clearly, make the results more palatable and provide a more insightful visualization. The scaling results are also more directly aligned with the ground truth of the protocol and MATCH flowcharts.

In the COPS analyses, we further find clusters of modules within each of the protocol facets. We used a classification that is also based on ESPCs to further investigate the within-protocol clusterings. We find that in the depression protocol, the usage of modules is diverse with respect to their ESPCs. This is less so in the anxiety and conduct protocols, where we find homogeneous ESPC clusters. It appears that clinicians' usage of modules is

diverse with respect to ESPCs if the primary diagnosis is one of depression. For conduct and anxiety problems, the variability in which ESPCs are used is lower and we find relatively homogeneous ESPC usage when looking at clusters of at least three modules. This may be because these ESPCs are seen as especially relevant or helpful in these two problem areas by the clinicians. It may also mean that it is a design property of the standard course of modules for anxiety and conduct to have modules of the same ESPCs occur together frequently, whereas this is not the case for depression. If a higher diversity of ESPCs per protocol is seen as better, then a revised version of MATCH-ADTC could take that into account. Further research may address these questions suggested by the exploratory analysis.

By using different stresses in COPS, we were able to highlight different aspects in the data. For example, multiscale and s-stress clearly indicate the "Booster" module as an outlier. The "Booster" module is one that is used with conduct problems and happens weeks after the other modules are already finished. In most scalings, "GettingAcquainted" and the relaxation modules are scaled close to each other, which could mean that clinicians use relaxation modules in a more general sense than as one belonging to the depression protocol; indeed, in most COPS scaling, we could classify the relaxation modules as GEN and still obtain a perfect faceting. Further research might look into the usage of the relaxation modules and how they might be best used in MATCH.

For the method itself, we believe that this article showcased the potential that COPS has as a method for multivariate data exploration in a practical setting. Even though we did a purely data-driven analysis, using COPS gave us a number of questions and hypotheses that are potentially interesting topics for further research around MATCH-ADTC and its usage and we invite interested readers to follow-up on them. Overall, we believe this article highlights the versatility and flexibility of the COPS framework for this type of analyses. Since standard MDS models are a special case of COPS, the COPS framework expands the possibilities practitioners have to analyze and explore their data with scaling and ordination techniques. Practitioners who are already familiar with using MDS can easily incorporate the COPS framework and the additional flexibility it offers in their daily work. In the best sense of modern exploratory data analysis, COPS provides a way to look at what the data seem to say. As our analyses showed, COPS can uncover underlying structures and isolate important aspects in the data, some of which are expected, but some are also surprising. We thus believe COPS to have tremendous potential for exploring data based on dissimilarities.

## 5. Limitations

This study is not without limitations. First, in general, the results of proximity scaling depend on the choice of dissimilarity measure and there are a myriad to choose from. We believe that it is a good idea to think deeply about what properties a dissimilarity measure should have for the data at hand. It often takes a lot of thought and trial-and-error with different dissimilarity measures to find one that works well for an application, as was the case for the present study.

Second, the optimization in COPS is very difficult due to the combination of the smooth and non-convex stress measures with the OPTICS Cordillera that captures discrete structures. Therefore, the typical objective function in COPS can have multiple local optima, may not be smooth and can show discontinuities [1]. Optimization therefore has to rely on heuristics that have no guarantees to converge to a global maximum and we recommend running the optimization process for a number of different starting configurations and choose the one that is smallest over these runs (we used five different starting configurations). While this still does not guarantee that we found the global optimum, at least we have chosen the lowest local optimum we encountered.

Third, all the methods we use have a high number of gauges to tune and refine the results. While we do believe that this flexibility is good in an exploratory setting (cf. [42]), it leaves many decisions to be made by the analyst and this can feel overwhelming. We typically approach this in two ways: We start with using defaults and systematically vary

the parameters that have the most influence on the results. For example, for OPTICS, the OC and COPS, varying the $k$ (minimum points per cluster) is something we do routinely, as is using different $\lambda, \kappa$ in COPS. Additionally, $d_{max}$ often needs to be adjusted to get a good balance between efficiency and robustness. Other important choices are the type of PS loss we want to use with COPS, and setting the $v_1$ and $v_2$; the latter has a bearing on how easy the objective is to optimize, with $v_2 \to 0$ making optimization easier. Similarly, in SVM the choice of kernel can also have a strong effect on the result as well as the choice of cost constraints (which we chose by cross-validation). For all of these parameters, it is worthwhile to explore their effect for a given data set and to try out different values and/or systematically vary them. There are also less important parameters, such as the $q$ or $\epsilon$ in OPTICS, the OC or COPS, that can just be set and typically need not be varied. In any case, and with all these methods, we recommend thinking deeply about the effects of these gauges and which ones should be turned. It is, however, difficult to know what the "best" choice would be.

Fourth, we operated in an exploratory setting as opposed to a confirmatory one. We thus aim at creating many different results and this includes some that may be dead ends. As Tukey states for exploratory data analysis ([42] p. 3): "To fail to collect all appearances because some–or even most–are only accidents would, however, be gross misfeasance [...]". In such a setting, we therefore need to try out a lot of parameters and see what changes, and the many different gauges help in that regard. We can then look at what changes: If changes in parameters lead to large changes in results, then we view this as an opportunity for exploration—every parameter has a defined effect and the results can be interpreted in light of that. For example, if one has two results with different values for $\sigma_{COPS}$ obtained from different starting configurations, but otherwise same parameter setup, then at least one of them was trapped in a local minimum. If two results use different $k$, $d_{max}$ or $\epsilon$, then this allows the exploration of the configuration for different cluster and clusteredness definitions. If two results differ because of different stress transformations that are used (e.g., interval vs. multiscale), then this allows exploration of how different mappings of dissimilarities to fitted distances play out. It is in this spirit that we suggest one uses the methods we presented. However, it can also be that any number of these results are just "accidents", so whatever substantive insights are gained that way should be followed up by confirmatory approaches.

Fourth, for the substantive results, we found somewhat ironically what we should have suspected all along, namely that the usage of the MATCH modules matches in large parts the guidance in the instruction manual and flowcharts. While our results do suggest intra-modular clusterings that may be worth exploring further, the results were not as illuminating as we hoped beyond what experts of MATCH already knew. Thus, it may be that MATCH was not the best choice for exploring the usage of modular therapies in general as the course of MATCH modules is highly standardized and most clinicians seem to follow it. A promising future direction could be to look at the usage of a modular psychotherapy whose guidance is less standardized.

## 6. Computational Details

COPS models as illustrated can be fitted with the R package **cops**. For OPTICS, we used **dbscan** [16]. SVMs were fitted with **e1071** [19] and SVM-MDS can be plotted with **smacof** [18]. For plotting in this article, we used **ggplot2** [43] with **ggthemes** [44] and **ggrepel** [45] for label placement and **ggplotify** [46] was used to turn base plots into ggplots. Colors were provided by **colorspace** [47]. To arrange the plots, we used **gridExtra** [48].

**Supplementary Materials:** The following supporting information can be downloaded at: https://www.mdpi.com/article/10.3390/psych5020020/s1, Code S1: cops-psych.R.

**Author Contributions:** Conceptualization, T.R., K.V.-C. and P.M.; data curation, T.R. and K.V.-C.; formal analysis, T.R., K.V.-C., G.B. and P.M.; investigation, K.V.-C.; methodology, T.R., K.V.-C., G.B. and P.M.; project administration, T.R. and P.M.; resources, K.V.-C.; software, T.R. and P.M.; supervision,

P.M.; validation, T.R.; visualization, T.R. and G.B.; writing—original draft, T.R., K.V.-C. and P.M.; writing—review and editing, T.R., K.V.-C., G.B. and P.M. All authors have read and agreed to the published version of the manuscript.

**Funding:** This research received no external funding.

**Institutional Review Board Statement:** The studies from which these data were drawn were conducted in accordance with the Declaration of Helsinki, and approved by the Institutional Review Board of Harvard University or Judge Baker Children' Center at Harvard Medical School (protocol codes NCT02877875, NCT03153904, and NCT03112304 and dates of approval 3.10.2014, 24.10.2013, and 20.05.2011).

**Informed Consent Statement:** Informed consent from all guardians and assent from all minors was obtained for this study.

**Data Availability Statement:** Restrictions apply to the availability of the raw data. The raw data were obtained from the Lab for Youth Mental Health at Harvard University and are available from Katherine Venturo-Conerly with the permission of the Lab for Youth Mental Health. The generated data analyzed in this study (the $\phi$-distance matrix) is publicly available in **cops** version 1.3-5 or higher. The most recent version of the package can be found here: https://r-forge.r-project.org/R/?group_id=2037 (accessed on 12 April 2023)

**Conflicts of Interest:** The authors declare no conflict of interest.

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
