# Peer review of "COPS in Action: Exploring Structure in the Usage of the Youth Psychotherapy MATCH"

_psych, doi:10.3390/psych5020020_

Round 1

Reviewer 1 Report

Thanks for the opportunity to review the manuscript titled „COPS in Action: Exploring Structure in the Usage of the Modular Approach to Therapy for Children (MATCH)“ that serves as a tutorial for applied researchers who are interested in getting familiar with modern approaches of proximity scaling techniques, in particular, Cluster Optimized Proximity Scaling (COPS). The paper introduces the basic ideas of these approaches and demonstrates how they can be applied based on the R package cops with an empirical example on the usage of MATCH modules by youth psychotherapists.

I enjoyed reading the well-written paper and I could follow the lines of argumentation quite well even though I am not very familiar with (advanced) proximity scaling techniques. The application of the method(s) and details about the choice of different parameter settings are richly illustrated by the example data with non-technical interpretations of the results. Therefore, I can clearly recommend this paper for publication in Psych. Below are some minor issues that might be addressed and a few typos.

-        Equation 4: I wondered  if the denominator (x_il + x_jl) can never be zero.

-        Figures 9-10: Is it possible to increase the readability by specific settings in the respective R package.

-        p. 22: The application of the SVM might be explained a little bit more in detail for readers who are not familiar with this technique (e.g., what does “with a radial kernel and a degree of 3” mean?).

-        I wondered if – besides the example data - there are some data constellations which typically lead to estimation problems or that are quite sensitive to the choice of parameters. For instance, are there some rough guidelines for applied researchers on the minimum sample sizes for different variable types and number of variables? What should be done or what does it mean for the interpretation of the results, if large changes in the overall picture result from different parameter settings?

Typos

-        p. 2, line 68: “we give an”

-        p. 22, line 589: “ESPCS” => “ESPCs”

Author Response

I enjoyed reading the well-written paper and I could follow the lines of argumentation quite well even though I am not very familiar with (advanced) proximity scaling techniques. The application of the method(s) and details about the choice of different parameter settings are richly illustrated by the example data with non-technical interpretations of the results. Therefore, I can clearly recommend this paper for publication in Psych. Below are some minor issues that might be addressed and a few typos.

Thank you for your time, the kind words and the helpful feedback. We address all the points below.  

-        Equation 4: I wondered  if the denominator (x_il + x_jl) can never be zero.

Good point - it can be zero if the phi distance of two zero vectors (so only zero elements) is calculated. We now mention that in the article and that we expect each vector to have at least one non-zero element. 

-        Figures 9-10: Is it possible to increase the readability by specific settings in the respective R package.

We enlarged Figure 10 by 10%. For Figure 9 it is not possible to have the labels not be overplotted; however the important info here are the facets/regions; the configurations in Figure 1 and Figure 8 are the same ones. We added that to the text. 

-        p. 22: The application of the SVM might be explained a little bit more in detail for readers who are not familiar with this technique (e.g., what does “with a radial kernel and a degree of 3” mean?).

We added a bit more explanation in the method section. Also, "degree of 3" was wrong and we removed it. 

-        I wondered if – besides the example data - there are some data constellations which typically lead to estimation problems or that are quite sensitive to the choice of parameters. For instance, are there some rough guidelines for applied researchers on the minimum sample sizes for different variable types and number of variables? What should be done or what does it mean for the interpretation of the results, if large changes in the overall picture result from different parameter settings?

These are good questions. The optimization in COPS is very difficult and the function to optimize can at present only be tackled with heuristics. In practice this can lead to estimation difficulties for any data size. In terms of parameters, most effective are the weights v_1 and v_2 - the smaller v_2 is, the more "well-behaved" the estimation problems becomes (as optimizing the Cordillera as a function of X is nasty). Well-behaved in terms of becoming a pure stress optimization problem, which is still hard due to its non-convex nature. We added a few sentence along these lines and mention it in the new limitations section. Another aspect is that it is prudent to use different starting configurations and then choose the solution with the minimum obtained \sigma_{cops}. We also added that to the methods section. If changes in parameters lead to large changes in results, then this is an opportunity for exploration - as every parameter has a defined effect the results can be interpreted in light of that. For example, if one has two results with the same parameter setup but different stresses, then at least one of them got trapped in a local minimum. If two results use different minpts, dmax or epsilon then this allows the exploration of the configuration for different cluster and clusteredness definitions. If two results differ because of different stress transformations that are used (e.g. interval vs multiscale) then this allows exploration of how different mappings of dissimilarities to fitted distances play out. Opposed to confirmatory settings, it is important to note that in exploratory data analysis we aim at creating many different results including those that are dead ends; as Tukey (1977, p.3) states: "To fail to collect all appearances because some--or even most--are only accidents would, however, be gross misfeasance."  We added that to the new limitations section.

Typos

-        p. 2, line 68: “we give an”

-        p. 22, line 589: “ESPCS” => “ESPCs”

Thanks, corrected.

Reviewer 2 Report

Reviewer's summary after reading the manuscript:

The present manuscript serves as an introductory exposition to Cluster Optimized Proximity Scaling (COPS), intended for professionals in the field, and also functions as a didactic guide on the application of the associated R software package, cops. The COPS methodology is a form of multidimensional scaling (MDS) that endeavors to generate a clustered configuration that accurately reflects multivariate dissimilarities. The statement implies that the mentioned approach encompasses a majority of prevalent MDS versions as particular instances. This study showcases the concepts, application, adaptability, and multifunctionality of the methodology and software through an examination of the utilization of MATCH youth psychotherapy modules by practitioners in real-world settings. The COPS analyses are complemented by density-based hierarchical clustering in the original space and faceting with support vector machines. The utilization of COPS for scaling yields a rational and perceptive spatial configuration of the modules, facilitates the straightforward recognition of module clusters, and presents distinct aspects of modules that correspond to the MATCH protocols. The efficacy of COPS surpasses that of conventional MDS and clustering methodologies.

------------------------------------------------

Dear authors, thank you for your manuscript. I enjoyed reading it. Presented are some suggestions to improve it:

(1) Please consider modifying the title of the manuscript to include the words "psychotherapy" so that it would be easier for potential readers to find your study. Please also include "psychotherapy" inside the keywords list.

(2) Please include a "Limitations" section to discuss what were the challenges faced, and how your team overcame those challenges. This would be very beneficial to the readers as they would be able to learn from your expert knowledge.

(3) To improve the impact and readership of your manuscript, the authors need to clearly articulate in the Abstract and in the Introduction sections about the uniqueness or novelty of this article, and why or how it is different from other similar articles. Can the authors please kindly elaborate more about how this study is relevant to "psychology" since it was submitted for publication in the journal entitled "Psych"?

(4) Please substantially expand your review work, and cite more of the journal papers published.

(5) All of the references cited are not yet properly formatted according to MDPI's guidelines. For example, all the journal papers cited are not yet sorted in A-Z alphabetical order. For the references, instead of formatting "by-hand", please kindly consider using the free Zotero software (https://www.zotero.org/), and select "Multidisciplinary Digital Publishing Institute" as the citation format, since there are currently 46 citations in your manuscript, and there may probably be more once you have revised the manuscript.

Thank you.

Author Response

Dear authors, thank you for your manuscript. I enjoyed reading it. Presented are some suggestions to improve it:

Thank you for the kind words and the suggestions. We followed them when applicable. Our answers are in italics.

(1) Please consider modifying the title of the manuscript to include the words "psychotherapy" so that it would be easier for potential readers to find your study. Please also include "psychotherapy" inside the keywords list.

We changed the title to "COPS in Action: Exploring Structure in the Usage of the Youth Psychotherapy MATCH". We also added psychotherapy to the keyword list.

2) Please include a "Limitations" section to discuss what were the challenges faced, and how your team overcame those challenges. This would be very beneficial to the readers as they would be able to learn from your expert knowledge.

We added that (Section 5).

(3) To improve the impact and readership of your manuscript, the authors need to clearly articulate in the Abstract and in the Introduction sections about the uniqueness or novelty of this article, and why or how it is different from other similar articles. Can the authors please kindly elaborate more about how this study is relevant to "psychology" since it was submitted for publication in the journal entitled "Psych"?

We added sentences on the novelty and relevance of our study to the introduction and the abstract. We believe with this being the first empirical study into the usage of modular youth psychotherapy the novelty of the study is given. Together with our substantive results based on youth psychotherapy data, we think the relevance for Psychology is established.  Our topic also fits neatly within the scope of the special issue on "Computational Aspects and Software in Psychometrics".

4) Please substantially expand your review work, and cite more of the journal papers published.

Respectfully, we don't think this is good practice. All the references that were important for our study have already been included. 

5) All of the references cited are not yet properly formatted according to MDPI's guidelines. For example, all the journal papers cited are not yet sorted in A-Z alphabetical order. For the references, instead of formatting "by-hand", please kindly consider using the free Zotero software (https://www.zotero.org/), and select "Multidisciplinary Digital Publishing Institute" as the citation format, since there are currently 46 citations in your manuscript, and there may probably be more once you have revised the manuscript.

We added doi's to the references we could find them for as well as capitalized titles and journals in line with the style guide. Beyond that we're not sure why the references are not matching MDPI's format as we manage the references in Bibtex and compiled them using the official LaTeX and bibtex template of MDPI with option psych. Therefore no typesetting on our part has been done and it is not possible for us to change the formatting further.